# Chemical characterization of Punta de Fuencaliente CO₂ enriched system (La Palma Island, NE Atlantic Ocean): a new natural laboratory for ocean acidification studies

Sara González-Delgado[1], David González-Santana[2,3], Magdalena Santana-Casiano[2], Melchor González-Dávila[2], Celso A. Hernández[1], Carlos Sangil[1], José Carlos Hernández[1].

[1] Departamento de Biología Animal, Edafología y Geología, Facultad de Ciencias, Universidad de La Laguna, Canary Islands, Spain.
[2] Instituto de Oceanografía y Cambio Global, IOCAG-ULPGC. Universidad de Las Palmas de Gran Canaria, Canary Islands, Spain.
[3] Univ Brest, CNRS, IRD, Ifremer, LEMAR, F-29280 Plouzane, France.

*Corresponding author:* Sara González-Delgado (sgonzald@ull.edu.es)

**Abstract.** We present a new natural carbon dioxide ($CO_2$) system located off the southern coast of La Palma Island (Canary Islands, Spain). Like $CO_2$ seeps, these $CO_2$ submarine groundwater discharges (SGD) can be used as an analogue to study the effects of ocean acidification (OA) on the marine realm. With this aim, we present the chemical characterization of the area, describing the carbon system dynamics, by measuring pH, $A_T$ and $C_T$, and calculating $\Omega$ aragonite and calcite. Our explorations on the area have found several emission points with similar chemical features. Here, the $C_T$ varies from 2120.10 to 10784.84 μmol kg-1, $A_T$ from 2415.20 to 10817.12 μmol kg-1, pH from 7.12 to 8.07, $\Omega$aragonite from 0.71 to 4.15 and $\Omega$calcite from 1.09 to 6.49 units. Also, the $CO_2$ emission flux varies between 2.8 to 28 kg $CO_2$ d-1, becoming a significant source of carbon. These $CO_2$ emissions, which are of volcanic origin, acidify the brackish groundwater that is discharged into the coast and alter the local seawater chemistry. Although this kind of acidified system is not a perfect image of future oceans, this area of La Palma island is an exceptional spot to perform studies aimed to understand the effect of different levels of OA on the functioning of marine ecosystems. These studies can then be used to comprehend how life has persisted through past Eras, with higher atmospheric $CO_2$, or to predict the consequences of present fossil fuel usage on the marine ecosystem of the future oceans.

**Keywords.** Volcanic, hydrothermal, brackish water discharge, groundwater, ocean acidification, ocean chemistry.

## 1 Introduction

Since the last decade, marine systems with natural carbon dioxide ($CO_2$) sources have been used as analogous of the acidified future oceans to understand its effects on organisms and marine ecosystems functioning (IPCC, 2014; Hall – Spencer et al., 2008; Foo et al., 2018; González-Delgado and Hernández, 2018). These areas are characterized by an extra-$CO_2$ input from volcanic (normally called $CO_2$ seeps), karstic or biological sources or originate from upwelling (González-Delgado and Hernández, 2018). Due to its origin, $CO_2$ vent systems are very common and can be found all over the world from mid-oceanic ridges to oceanic island and intra-plate magmatism (Dando et al., 1999; Tarasov et al., 2005). In general, the vent systems have emissions in the form of bubbles which are 90-99% $CO_2$.. The most notable features of these acidified systems are the fluctuation of pH, the aragonite and calcite saturation states ($\Omega$) (declining between 1 and 3) and dissolved inorganic carbon (DIC) which increases up to 3.2 mol C m-3 (González-Delgado and Hernández, 2018).

Moreover, there are marine shallow areas affected by $CO_2$ gas diffusive emissions through submarine groundwater discharges (SGD) that acidify the surrounding waters (Hall-Spencer et al., 2008).

Numerous advances in ocean acidification (OA) studies have been achieved using these systems, such as acidification effect on ecology interaction (e.g. Nagelkerken et al., 2016), physiological (e.g. Migliaccio et al., 2019) and genetic adaptations (e.g. Olivé et al., 2017). Nowadays, it is possible to better understand the direct and indirect effects of OA in marine environments

due to these acidified systems, for instance we now know that OA related changes will reflect in the services that ecosystems provide to us (Hall-Spencer and Harvey, 2019). Acidified systems can be also used to look back into the past of the Earth, and to study how early life could have originated on the planet (Martin et al., 2008). Understanding how life has adapted in the past acidified Eras, can be extremely useful to understand how current life will change in the expected future (Gattuso et al., 1998).

The Canary Islands, located in the North-Eastern Atlantic Ocean, are an oceanic volcanic archipelago formed by numerous hotspot island chains (Carracedo et al., 2001). The youngest islands are El Hierro with 1.1 million years and La Palma with an age of 2 million years (Carracedo et al., 2001). These islands are located to the west of the archipelago and it is where the last historical eruptions have taken place. The last two were the Teneguía volcano in La Palma in 1971, and the Tagoro volcano in El Hierro in 2011 (Padrón et al., 2015; Santana-Casiano et al., 2016).

Currently, in the historic volcanic area in the south of La Palma (Cumbre Vieja volcano complex), there is a continuous degassing of $CO_2$ (Carracedo et al., 2001; Padrón et al., 2015). Correspondingly, on the nearby shore, $CO_2$ emissions have been detected recently in two different locations: Las Cabras site (Hernandez et al., 2016) and Punta de Fuencaliente, which has already been used for OA ecological studies (Pérez, 2017; Viotti et al., 2019). However, in these works only the pH and $pCO_2$ were measured, in localised points where certain samples were taken.

The local name "Fuencaliente", which translates into hot-springs, refers to the thermal fresh waters that emerge at the coast. Before the conquest of the islands in 1492, its waters were used by locals for its healing properties, and after that by visitors from all over the world (Soler, 2007). However, these *thermas* were buried by the eruption of the San Antonio volcano in the 17th Century. These thermal waters have been so famous and important for Fuencaliente people, that there was an engineering project to dig up these special waters (Soler, 2007). The brackish water features measured by Soler (2007) showed high

concentrations of bicarbonate ($HCO_3^-$), sulphate ($SO_4^{2-}$), chloride ($Cl^-$), that together with high temperatures (almost 50 ºC) confirmed the influence of internal magmatic activity. Nearby, there are brackish lagoons located in the innermost part of Echentive beach, about 200 m from the coastline with diameters of 30 m and depths of up to 4 m (Fig. 1). Measures of oxygen isotope ($\delta$ $^{18}$O SMOW) (Calvet et al., 2003) suggest a slight dilution of the seawater in the lagoons by inland brackish groundwater flowing into it. This indicates that in the system there are groundwater discharges, which probably comes from

the thermal waters studied by Soler (2007).

In the last two decades, an increasing number of studies underlined the importance of SGD (Jeandel, 2016). SGD is an essential but poorly recognized pathway of material transport to the marine environment (Szymczycha et al., 2014). The term SGD includes the discharge of fresh groundwater to coastal seas to which recirculation of seawater often contributes (Burnett et al., 2006, Charette et al., 2016). For issues related to oceanography, the term is restricted to fluid circulation through continental

shelf sediments with emphasis on the coastal zone (Burnett et al., 2006, Jeandel, 2016). One aspect that has yet not been considered is, what occurs in areas where SGD is enriched by the emissions of recent volcanism or by hydrothermal activity?. In this case, these discharges can also act as sources of gases and hydrothermal emission compounds to the ocean and become points of emission of $CO_2$ that contribute to the OA. However, shallow coastal beaches and intertidal lagoons are highly dynamic systems controlled by physical processes and subjected to marine and continental influences. Processes as the tide or

the submarine groundwater discharges produce higher ranges of variation in physical and chemical parameters than in the open ocean water and could provide a natural environment for laboratory studies.

Hence, and with the purpose of using the Punta de Fuencaliente area as a natural acidified laboratory, an accurate physical and chemical characterization of this area is presented in this study. The main objectives were (1) determining the area affected by

the emissions and detecting new emissions points for replication studies, (2) characterize the ocean chemistry of the area, and (3) to confirm the volcanic origin of the acidification.

## 2 Material and methods

### 2.1 Study area

The physical-chemical parameters were sampled across the south of La Palma island, located in the west of the Canary Islands (North-Eastern Atlantic Ocean) (Fig. 1a, supplementary material 1). The sampling took places between 0 and 2 m depth, at three different times (March 2018, December 2018 and June 2019) and during low and high tide when it was necessary to assess the continuity of the natural emissions (Fig. 1b, appendix A). Following the previous studies in the area (Hernández et al., 2016; Pérez, 2017; González-Delgado et al., 2018a; González-Delgado et al., 2018b; Hernández et al., 2018; Viotti et al., 2019), a sampling network was created for the first time. It is formed by seven sites: Playa del Faro, Los Porretos and surroundings (that together with Las Cabras site, they are known as Punta de Fuencaliente system or PFS), Playa Echentive and the two Echentive lagoons (Fig. 1c).

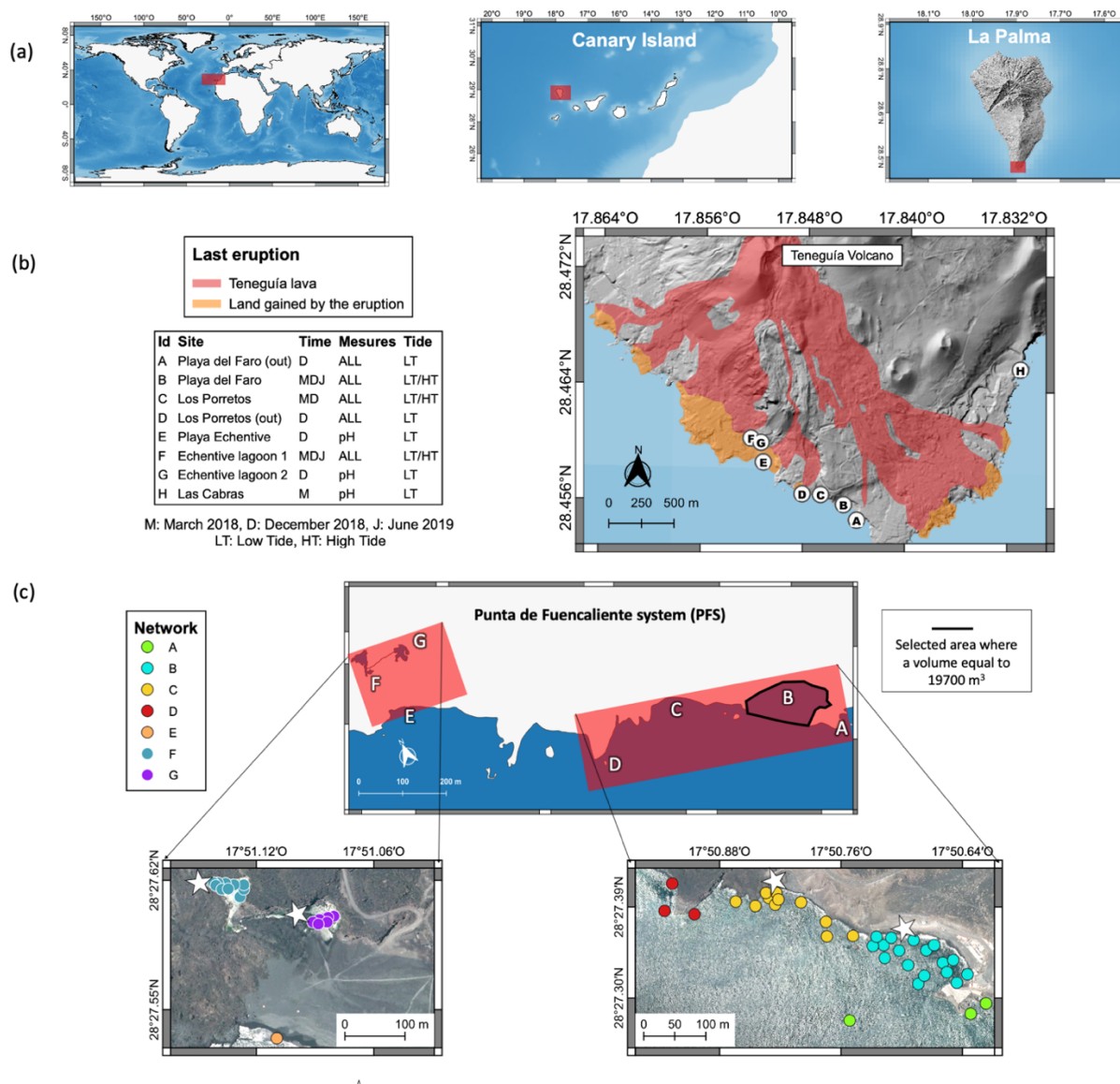

Figure 1: (a) Location of study are on North-Eastern Atlantic Ocean, in the west of the Canary Island, in the South of La Palma Island. (b) Location of the seven sampling sites (A-G) in the south of the island of La Palma. The location of Punta Las Cabras (H)

considered in Hernández et al. (2016) is also included. The area covered by the last volcanic eruption, Teneguía volcano, is also indicated. (c) Location of sampling network performed in this work around the Punta de Fuencaliente system (PFS) and the selected area where the volume was calculated to $CO_2$ flux calculation. The stars are included as a help to better interpret and locate the interpolation graphics from Figure 2-4. The map and image base layers used are distributed under public domain (https://www.grafcan.es/).

Scuba diving was used for sampling all bottles except in the reference station out of Playa del Faro (Fig. 1c A), where a CTD rosette was used. For the scuba sampling, the bottle was previously rinsed three times at the sampling location and then the bottle was immersed with the mouth down and turned at 1 m depth for sampling. Samples were poisoned with 100 µl of saturated $HgCl_2$ solution, sealed, kept at darkness, and analysed on the lab. In March 2018, this was done the same day while in December it was done two days later. For pH 100 ml borosilicate glass bottles were filled with seawater.

## 2.2 Carbon dioxide system parameters

In March and December of 2018, the total dissolved inorganic carbon concentration ($C_T$), total alkalinity ($A_T$), pH, salinity and temperature were measured, whilst in June 2019 only the pH and temperature were measured. Total alkalinity and $C_T$ were respectively determined by potentiometric and coulometric methods, using a VINDTA3C system (Mintrop et al., 2000). The calibrations were made using certified reference material batch #163 (González-Dávila et al., 2007). The pH was measured at a constant temperature of 25 °C in less than one hour from sampling, using an Orion pH meter with a combined Orion glass electrode ($pH_{T,is}$). The calibration was performed on the total seawater scale using a TRIS artificial seawater buffer (salinity 35) according to the Guide to best practices for ocean $CO_2$ measurements (Dickson et al., 2007, SOP 6a).

Salinity and temperature were measured *in situ* using a handheld conductivity meter (Hanna Instruments HI98192). Furthermore, 200 ml salinity bottles were measured on the laboratory in less than two days and using a high-precision Portasal salinometer, accurate to ±0.001. The pH under "*in situ*" conditions, the partial pressure of carbon dioxide (pCO2) and the saturation states of calcium carbonate forms (Ω aragonite and Ω calcite) were determined from $A_T$ and $C_T$ data using the CO2sys program (Pierrot et al., 2006).

Atmospheric $CO_2$ concentrations used for fluxes calculations were those measured at the Izaña station at the island of Tenerife (IZO site and available in the World Data Centre for Greenhouse Gases).

We used the linear interpolation method to represents the $A_T$, $C_T$, $pH_{T,is}$, Ω aragonite and calcite parameters measurement when anomalies were found.

## 3 Results

After extensive sampling throughout the south of La Palma, we detected four areas where natural enrichment of $CO_2$ groundwater emissions occur. These four areas, Las Cabras, La Playa del Faro, Los Porretos and the two Echentive Lagoons (Fig. 1b,c), correspond to areas that were not buried by the lava during the last eruption (Teneguía volcano 1971; Padrón et al., 2015, Fig. 1b). Las Cabras site was discarded in subsequent samplings due to the difficult access, the poor sea conditions and the small size of the area affected by the emissions (Hernández et al. 2016). In all cases, the anomalies were the highest during low tide (appendix B)

### 3.1 Temperature and salinity

Temperature and salinity in Playa del Faro and Los Porretos do not present major changes between the different time points (supplementary material 2). During March 2018, Playa del Faro had an average temperature of 19.00 ± 0.20 ºC with colder values of 18.70 ºC near shore, Los Porretos was not measured this time. In December 2018, both Playa del Faro and Los Porretos presented an average temperature of 21.50 ± 0.02 ºC. However, salinity values present a minor diminution from 37.05

to 36.51 in Playa del Faro and from 37.05 to 36.07 in Los Porretos (supplementary material 2). Both sites presented colder and slightly less saline water near the coast. Regarding the Echentive lagoons, only the biggest lagoon was measured, where the salinity varied from 31.00 to 32.00 units (supplementary material 2). The same lagoon presented warmer temperatures than the coastal waters during June 2019, $26.40 \pm 0.70$ ºC and $22.00 \pm 0.10$ ºC, respectively.

## 3.2 Carbon dioxide system parameters

In both studied shore areas of PFS (Playa del Faro and Los Porretos) the parameters of the carbon dioxide system, $pH_{T,is}$ (Fig. 2A, B), $A_T$, $C_T$ and $\Omega$ aragonite and calcite (Fig. 3, 4B) were strongly affected by the entrance of the SGD with less salinity.

### 3.2.1 Playa del Faro

In March 2018, the pH changed from 8.06 in offshore samples to 7.50 nearshore, reaching 7.16 and 7.13 during December 2018 and June 2019, respectively (Fig. 2A). Similarly, high $A_T$ and DIC were measured throughout Playa del Faro. In March 2018, the ocean data obtained in the furthest coast station of Playa del Faro reached typical values of 2132.13 µmol kg$^{-1}$ and 2418.38 µmol kg$^{-1}$, respectively for $C_T$ and $A_T$ (supplementary material 2). As we approached the shore, both factors increased to values that exceeded 3100 µmol kg$^{-1}$, following an inverse distribution observed with salinity, with an increase in the $C_T$:$A_T$ ratio close to 1:1, indicating an important contribution of bicarbonate along the area (Fig. 3a, b). In December 2018, the anomaly increased to over 3500 µmol kg$^{-1}$ in both parameters. As a direct consequence of the low pH values, although compensated by the high $C_T$, $A_T$ and dissolved calcium contents (determined by ICP-MS, data not presented), the calcite and aragonite saturation states were also affected. It was observed that the area nearest to the shore presented saturation values of calcite and aragonite that were below 1.50 (Fig. 3c, d).

During high tide, the anomalies almost disappeared, which means that the tide acts as a pressure plug of the flow of this water to the coastal area. Nevertheless, we still found a mild increase in $A_T$ and $C_T$ (reaching 2692.13 and 2512.35 µmol kg$^{-1}$ respectively) (Fig. 3a, b) and pH values of 7.75 - 7.85 in the sampling points closest to the coast (Fig. 2A).

### 3.2.2 Los Porretos

Los Porretos, is a continuation of Playa del Faro that is also affected by the SGD with high $C_T$ and low pH. This discharge was first observed during March 2018. The measured $C_T$ exceeded 3400 µmol kg$^{-1}$ and the $pH_{T,is}$ reached 7.25 at the emission station (Fig. 2B, Fig. 4B). In December, the sampling was repeated, observing that the most anomalous values occurred in the stations closest to the coast. The emission point presented $C_T$ concentrations of 3456.6 µmol kg$^{-1}$ (corresponding with carbon dioxide pressure values of 5200 µatm), pH values of 7.27, and 1.45 and 0.95 values of $\Omega$ calcite and aragonite, respectively (Fig. 2B, Fig. 4B, supplementary material 2).

In both beaches, the emission is acting as an important source of $CO_2$ into the atmosphere. In Playa del Faro, the partial pressures of $CO_2$ in surface waters reached up to 5000 µatm in low tide (the values in the atmosphere were between 405 and 410 µatm) (supplementary material 2). This produced high concentration gradients that combined with high-intensity winds characteristic of the area, produced $CO_2$ fluxes that can reach up to 1 mol m$^{-2}$ day$^{-1}$ (considering its main effects during low tide and Wanninkhof, 2014 for the gas transfer velocity coefficient) that amount up to 150 tons of $CO_2$ per year.

### 3.2.3 Echentive lagoons

The two lagoons at Playa Echentive (Fig. 1c) show the maximum anomalies in the south of La Palma. They presented low salinities and low pH, below 7.5 in all stations, and reaching 7.39 in the northwest during March 2018 (data only from the big lagoon) (Fig. 2C, D). Similarly, the $C_T$ was above 9700 µmol kg$^{-1}$, with comparable values for $A_T$ (Fig. 4Ca, b). These $C_T$ and $A_T$ concentration together with the low pH values counteracted the saturation states of calcite and aragonite that were,

respectively, never below 4.35 and 2.79 (Fig. 4Cc, d). Furthermore, when both lagoons were sampled during December 2018, similar concentrations were measured at low and high tide (Fig. 2C, D). The north-western part of the big lagoon presented the highest $C_T$ concentration (greater than 10000 µmol kg$^{-1}$) and the lowest pH reached 7.38 in low tide and 7.55 at high tide that coincided with a decrease in salinity and a mild temperature increase (Fig. 2C, D, supplementary material 2). The rest of the big lagoon remained at pH 7.58, like the small lagoon with a maximum pH of 7.63. However, the small lagoon presented a lower pH range, with a minimum of 7.50 in low tide and a maximum of 7.64 in high tide in the northern part (Fig. 2C, D). The water levels in both lagoons were tide dependent. The entry of salty marine water during high tide reduced the anomaly caused by the intrusion of lower salinity water rich in $C_T$ and $A_T$.

180

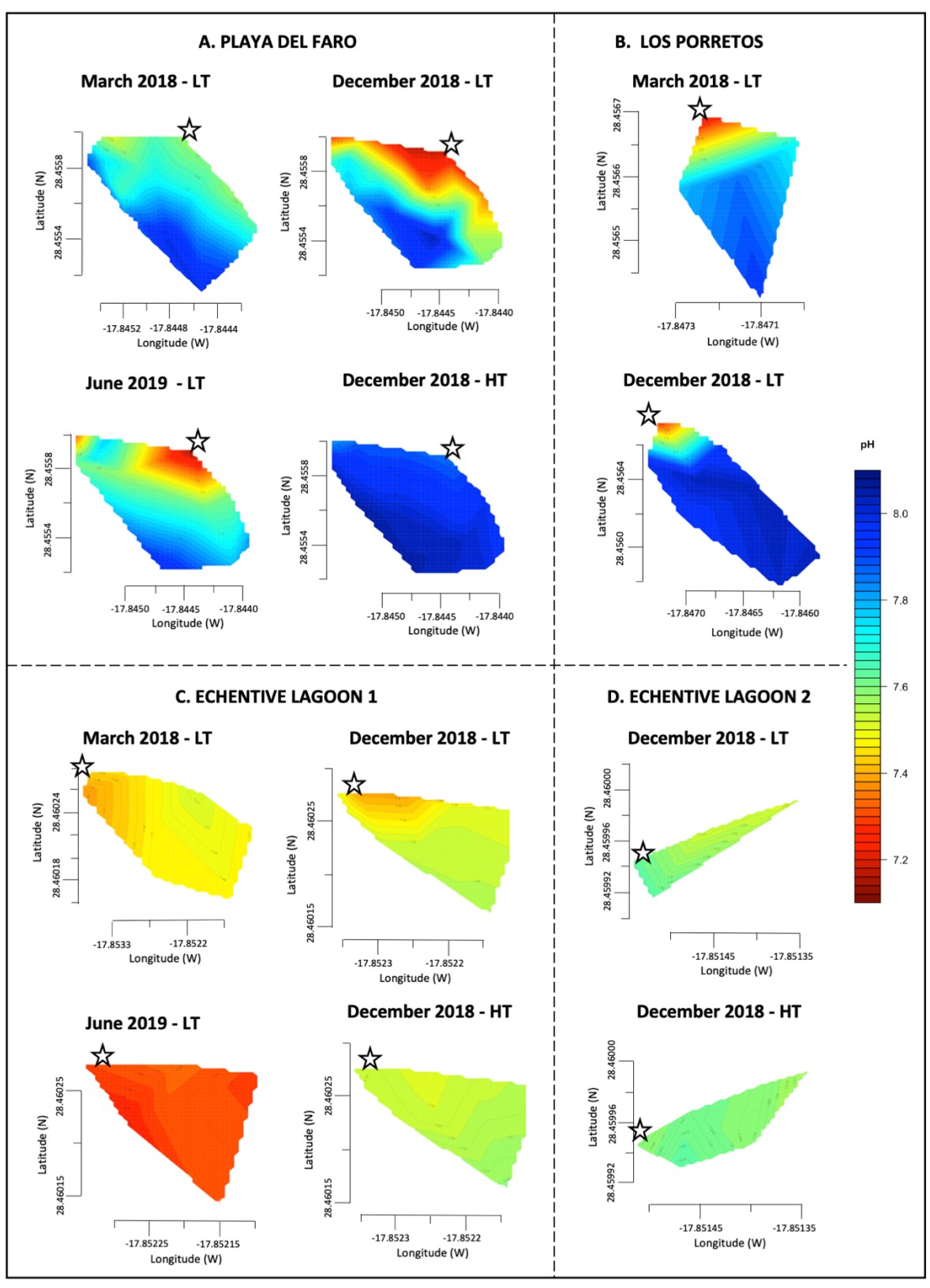

**Figure 2: Linear interpolation graphs of pH values that were collected in March 2018, December 2018 and June 2019 during low tide (LT) and high tide (HT) in Playa del Faro (A), Los Porretos (B), Echentive lagoon 1 (C) and Echentive lagoon 2 (D). The star symbol is the reference mark on the map in Figure 1c.**

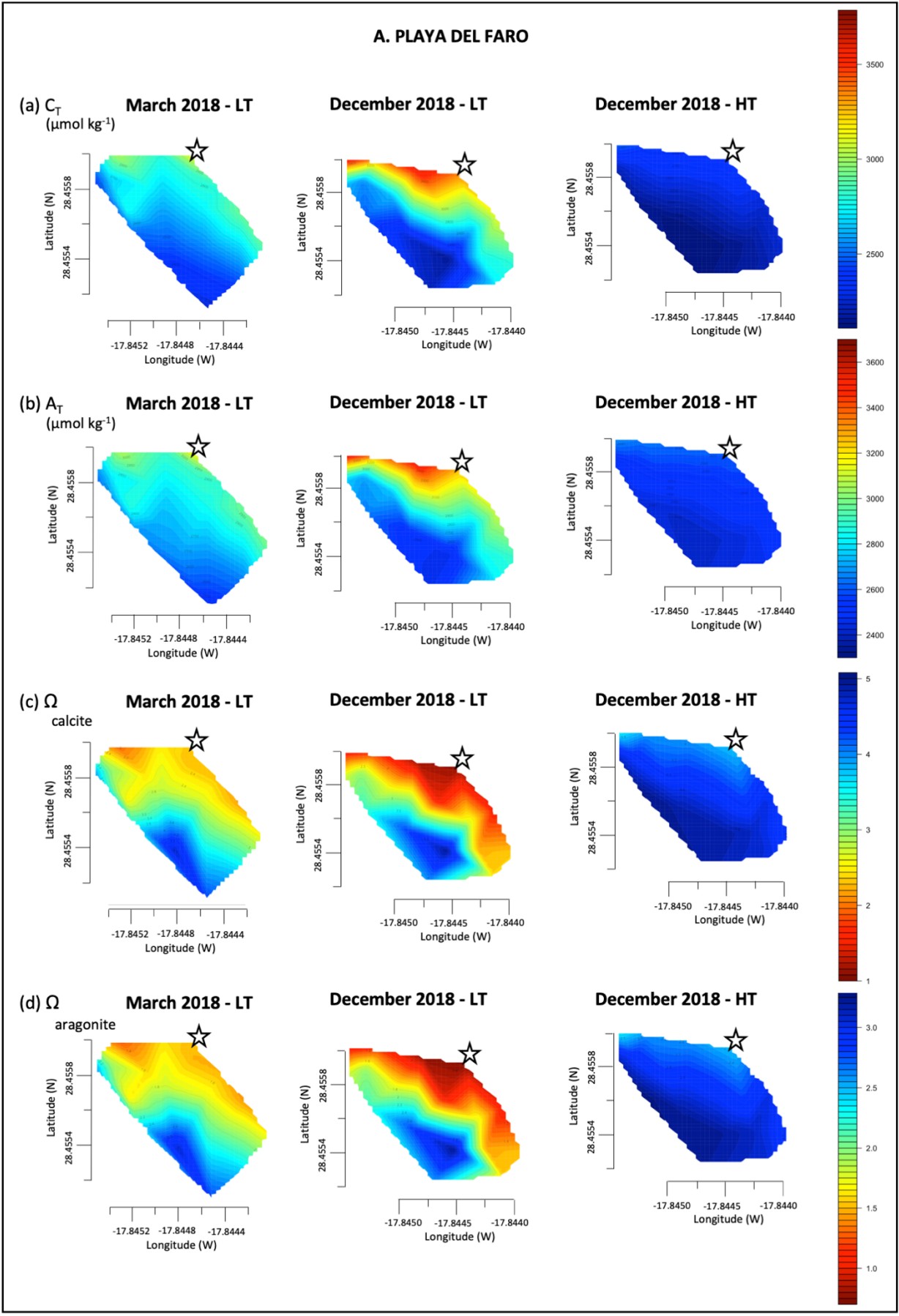

**Figure 3: Linear interpolation graphs of $C_T$ (a), $A_T$ (b), $\Omega_{calcite}$ (c) and $\Omega_{aragonite}$ (d) values during March 2018, December 2018 during low tide (LT) and high tide (HT) in Playa del Faro. The star symbol is the reference mark on the map in Figure 1c.**

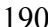

**Figure 4: Linear interpolation graphs of $C_T$ (a), $A_T$ (b), $\Omega_{calcite}$ (c) and $\Omega_{aragonite}$ (d) values during March 2018 and December 2018 during low tide (LT) in Los Porretos (B) and Echentive Lagoon 1 (C). The star symbol is the reference mark on the map in Figure 1c.**

## 3.3 CO$_2$ flux calculation

The CO$_2$ flux was calculated for Playa del Faro. We assumed two end members, the open ocean endmember and the SGD endmember. Soler (2007) discovered an aquifer near this area with brackish water (salinity = 30). Considering the bathymetry, the volume occupied by seawater was 19700 m$^3$. We also assumed that groundwater discharge only occurred at low tide. The average salinity changed from 36.93 (equivalent to 745.8 Tons of sea salt) at low tide to 37.02 at high tide (747.5 Tons of sea salt). The decrease in salinity at low tide could be accounted for by the emission of 57m$^3$ of brackish groundwater.

The brackish groundwater was also responsible for the A$_T$ and C$_T$ changes (Fig. 3a, b). Alkalinity increased by 219 µmol kg$^{-1}$ from high tide (2465 µmol kg$^{-1}$) to low tide (2684 µmol kg$^{-1}$). Considering 57 m$^3$ of brackish water, 4.40 kmol of alkalinity was required therefore, the brackish groundwater had an A$_T$ concentration of 76 mmol kg$^{-1}$. Similarly, the C$_T$ in the beach increased by 333 µmol kg$^{-1}$, from high tide (2190 µmol kg$^{-1}$) to low tide (2523 µmol kg$^{-1}$). The brackish water caused the increase of 6.7 kmol of inorganic carbon to the beach and therefore, had an endmember concentration of 116 mmol kg$^{-1}$. Considering the *in situ* temperature (20.67 ºC), the pH$_{T,is}$ decreased by 0.25 from 8.01 in hight tide to 7.76 in low tide. This meant that the acidity increased by 80%. This pH reduction meant that the water discharged on the beach had a pH of 5.57. The medium partial pressure of carbon dioxide for the area increased from 459 µatm at high tide to a value of 988 µatm at low tide. Considering an average wind speed at the beach of 7 m s$^{-1}$ (https://datosclima.es/Aemethistorico/Vientostad.php), La Playa del Faro acts as a strong source of CO$_2$, emitting 5.70 mmol CO$_2$ m$^{-2}$ d$^{-1}$ at high tide and increasing an order of magnitude at low tide (57 mmol CO$_2$ m$^{-2}$d$^{-1}$, Wanninkhof, 2014). Consequently, La Playa del Faro with its small area of only 0.01 km$^2$, is responsible for an atmospheric CO$_2$ emission flux varying between 2.80 kg CO$_2$ d$^{-1}$ to 28 kg CO$_2$ d$^{-1}$.

## 4 Discussion

### 4.1 The origin of the CO$_2$ submarine groundwater discharge

Although CO$_2$ emissions on Fuencaliente coast had already been detected (e.g. Hernández et al., 2016; Viotti et al., 2019), this is the first time that this naturally acidified system has been described chemically and physically. Previous works have focused on punctual questions; Hernández and collaborators (2016) published for the first time the presence of CO$_2$ SGD in Fuencaliente, specifically in Las Cabras beach. Later, in the thesis by Pérez (2017), as well as in the conference papers by González-Delgado et al. (2018a,b) and in the article by Viotti et al. (2019), new points of acidification were discovered on Playa del Faro and Los Porretos. However, in none of them was a chemical characterization of the whole area made as here. Our results reveal the continuous influence of brackish water discharge in the acidification process of Punta de Fuencaliente System (PFS), which had been missed before (Fig. 5). Similar to aerial remnant volcanic activity in La Palma that generates high CO$_2$-diffusive atmospheric concentration (Padrón et al., 2015), submarine remnant volcanic activity causes the acidification process found here, as indicated by the chemical composition of the groundwater analysed, which is less than 200 m from the coast (Soler, 2007). The activity of this SGD is comparable with other CO$_2$ vent and seep systems worldwide (references within González-Delgado and Hernández, 2018). Moreover, the presence of acidic water flow of La Palma also has a slight resemblance with the acidification phenomenon found in Mexico, originating from a karstic groundwater discharge (Crook et al., 2012). Furthermore, the highly alkalized and bicarbonate waters found in Echentive lagoons are an artefact of water discharge from the hydrothermally affected aquifers of the area (Soler, 2007), as found in Las Cañadas del Teide, in Tenerife (another island of the same archipelago) (Marrero et al., 2008).

In PFS there is a decrease in salinity due to brackish water discharges. Hence, there is a constant filtration of brackish acidified waters through high permeable volcanic rocks (Carracedo et al., 2001, Marrero et al., 2008), with chemical features due to underground volcanic activity, such as a 5.57 pH, a concentration of 76 mmol Kg$^{-1}$ of A$_T$ and 116 mmol Kg$^{-1}$ of C$_T$. However,

the effect on the surrounding seawaters depends upon tidal pressure and, more likely, other oceanic forces such as wind and waves (Moore, 2010; Mulligan et al., 2019).

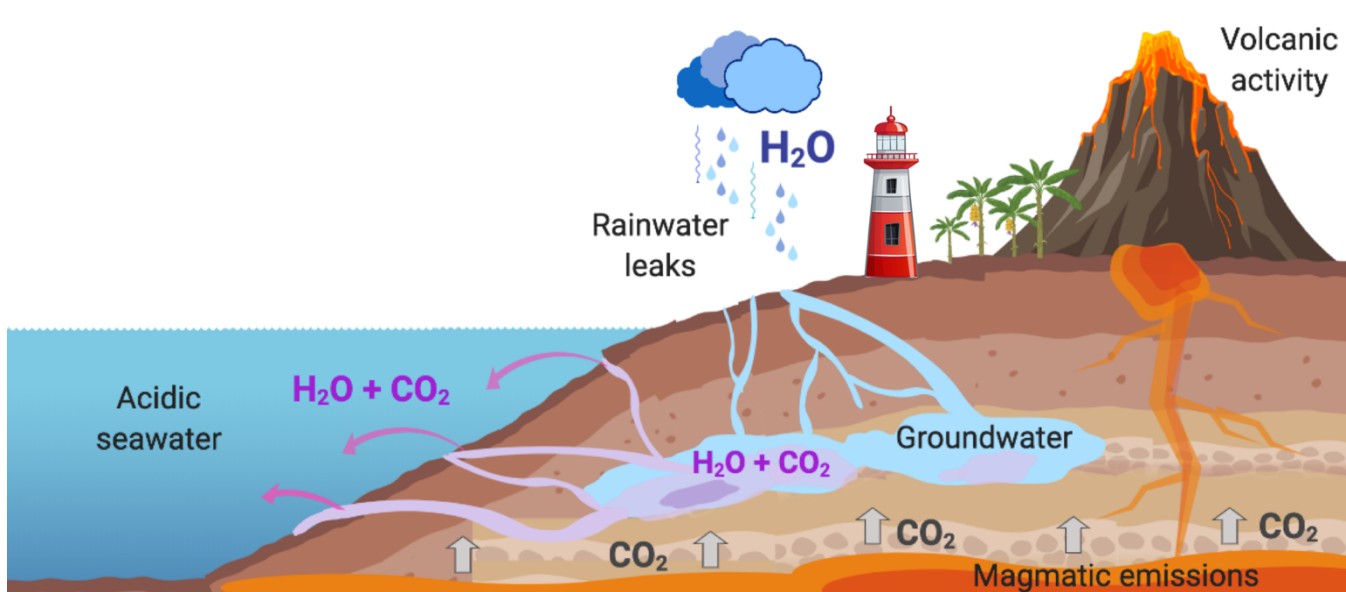

**Figure 5: Acidification process representation of Punta de Fuencaliente system (PFS) (made with Biorender).**

## 4.2 Alteration of the carbon chemistry system and implications for organism's assemblages

In the case of PFS, the water with lower salinity (36.79 - 36.45) and high concentration of $C_T$ and $A_T$ affect the surroundings, decreasing the seawater pH up to 0.8 and reducing the carbonates saturation state up to 1.1 of calcite and 0.7 of aragonite. This situation generates a carbon imbalance affecting carbonated organisms, especially those that precipitate aragonite on their calcareous structures (Kroeker et al., 2010). When the saturation values are below one, the formation of carbonates is not thermodynamically possible, although certain species require much higher saturation levels (Kroeker et al., 2010). The calcifying organisms that could live in these acidified areas may present weaker shells, skeletons and/or others solid structures, as we have recently observed in the mollusc *Phorcus sauciatus* (Viotti et al., 2019), as well as in the rest of calcifying organisms (Pérez, 2017). This excess of $CO_2$ has also modified the community composition and trophic structure, causing a loss of ecological and functional diversity on the benthic marine ecosystem (González-Delgado et al., in press).

In the case of Echentive lagoons, the anomaly is amplified due to a lower tidal influence and insulation. These acidified lagoons, which are at around 200 m distance from the coast (Fig. 1), have a salinity of 32 and $C_T$ and $A_T$ concentrations five times higher than normal ocean values. The $C_T$ and $A_T$ concentrations are so high that they compensate the decrease in pH with the content of carbonates in the water. These singular characteristics create a unique marine ecosystem. The environment is dominated by a biofilm of microorganisms predominantly microalgae, cyanobacteria and diatoms (Sangil et al., 2008), and probably other bacteria and fungi. Nonetheless, some marine invertebrates persist, such as the common errant polychaete *Eurythoe complanata* and the anemone *Actinia* sp. (Sangil et al., 2008). A more in-depth physiological study of these species could help us to better understand their adaptation process to these conditions and to give insights of what we might expect in future ocean acidification conditions, especially at the PFS area.

## 4.3 La Palma as a natural laboratory for marine research

The natural $CO_2$ gradients south of La Palma have been characterized from shore to offshore, varying for $C_T$ from 2120.10 to 3794.00 µmol $kg^{-1}$, for pH from 7.12 to 8.07, for $\Omega$aragonite from 0.71 to 3.28 and $\Omega$calcite from 1.09 to 5.02. This high local variability is in line with other acidified natural systems. For example, the $CO_2$ vent of Ischia (Italy) has pH levels from 6.07

to 8.17, $\Omega$aragonite from 0.07 to 4.28 and $\Omega$calcite from 0.11 to 6.40 (Hall-Spencer et al., 2008). The one from the island of Vulcano (Italy) has pH values between 6.80 and 8.20, $\Omega$aragonite from 1.49 to 4.65, and $\Omega$calcite from 2.28 to 7.00 (Boatta et al., 2013). Meanwhile the $CO_2$ seeps from Papua New Guinea have pH levels between 7.29 - 7.98, $\Omega$aragonite 1.2 - 3.4 and

$\Omega$calcite between 1.36 - 5.12 (Fabricius et al., 2011). That from Shikine island (Japan) have pH values between 6.80 and 8.10, $\Omega$aragonite from 0.20 to 2.22, and $\Omega$calcite from 0.30 to 3.45 (Agostini et al., 2015). Although these systems are far from being perfect predictors of the ocean future due to their chemical variability and physical limitations, they have proven to be important tools for the study of ocean acidification (Foo et al., 2018; González-Delgado and Hernández, 2018; Aiuppa et al., 2020). These natural acidified systems, such as Punta de Fuencaliente system (PFS), can be used as natural analogue of climate

change scenarios predicted by the IPCC (2014) (Fig. 6). Therefore PFS can be considered a very useful spot for large-scale and long-term adaptation experiments, as seen in other $CO_2$ systems (e.g. Ricevuto et al., 2014; Uthicke et al., 2019) . Moreover, the acidified system of La Palma highlight by the absence of bubbling, since the volcanic degasification takes place in the aquifers and not directly on the coast as in other acidified systems of volcanic origin (e.g. Hall-Spencer et al., 2008; Fabricius et al., 2011) (Fig. 5). This could give us new insights of the effect of acidification *in situ* avoiding the effects of

bubbling (González-Delgado and Hernández, 2018). Nevertheless, several caveats for future prediction experiments should be considered, as well as in other natural acidified systems, especially those related with increased alkalinity values in the submarine discharge.

First, there is a clear tidal influence, this is an important force that controls the acidified brackish water discharges. Although a fluctuation of the emission is observed, normal ocean conditions can occur for a short time, about 2-4 hours per day, during

high tide, and depending on the oceanic conditions (Viotti et al., 2019). The $pH_T$ is severely affected by the location, reaching down to ~7.2 in the emissions points, so a careful selection of the study sites is recommended, depending on the study objectives (Fig. 6). This tidal phenomenon has also been reported in other acidified natural systems such as Puerto Morelos in Mexico (Crook et al., 2012) and Ischia (Kerrison et al., 2011). However, the pH time fluctuation can be used to our advantage, as a daily and seasonal fluctuation of the pH is normal in coastal habitats environment (Hofmann et al., 2011). So, it could be

considered very useful to incorporate pH variability in ocean acidification studies as environmental fluctuations that can have a large impact on marine organisms (Hofmann et al., 2011).

Second, one of the most common concerns with $CO_2$ seeps and SGD areas is the presence of other gases or elements associated with volcanic emissions, such as nitrogen ($N_2$), mercury (Hg), methane ($CH_4$), etc. (e.g. Fabricius et al., 2011; Boatta et al., 2013; Aiuppa et al., 2020). Although there are no traces of the presence of harmful volcanic elements for marine organisms

such as methane or sulphates in the seawater of PFS (Hernández et al., 2016), there is an extra supply of different elements such us Mg that comes from groundwater (Soler, 2007). Groundwater has 10 times more magnesium than normal, but when mixed with seawater, the supply is considerably lower compared to $CO_2$. Nevertheless, Mg plays an important role in the calcification of marine organisms that have magnesite-calcite, such as echinoderms (Weber, 1969) and some bryozoans' species (Smith et al., 2006). Similarly, Hernández et al. (2016) found an increase in silicates in the nearby area of Las Cabras.

In these cases, Si could participate in the calcification of diatoms (Paasche, 1973) as well as in many sponges (Smith et al., 2013). The increase of these essential elements for certain calcifying species can allow their survival and growth in PFS while buffering the effects of acidification (Smith et al., 2016; Ma et al., 2009). Therefore, measurements of heavy metals and other elements in seawater should be considered in the following studies.

The high concentration of bicarbonate in the brackish waters also implies an extra contribution of alkalinity and carbonate that

can buffer the effect of acidification in the area, so it is necessary to take this into account when making predictions of the future. These values together with calcium content are especially important factors in the case of the saturation state for both calcite and aragonite, that shows high values for seawater with low pH values. Hence, despite the fact that we are dealing with a subtropical ecosystem, the values obtained in both saturation states are more similar to the predictions for a tropical ecosystem, such as the values found in Papua New Guinea seeps (Fabricius et al., 2011; IPCC, 2014).

Finally, the area is not very large and only one type of rocky benthic habitat, the most typical community of Canary island, is present at the PFS (Sangil et al., 2018). Therefore, all conclusions derived from this acidified system should be interpreted with caution and has local effects. Hence, it is crucial to establish a collaborative network of researchers who are working in other natural acidified systems worldwide to have a more realistic interpretation of future ocean scenarios.

The Echentive lagoons are an oversaturated carbonate system. Like hydrothermal alkalinity vents (Martin et al., 2008), it could 310 help us to understand the early life on Earth from the Precambrian, 4000 million years ago, when the atmosphere was rich in $CO_2$ (Kasting, 1993; Nakamura and Kato, 2004) (Fig. 6). These studies could allow us to disentangled adaptation and evolution of marine life to the changing carbonate conditions over time (Gattuso et al., 1998).

Additionally, and to our knowledge, it is the first time that a brackish water discharge altered by volcanic activity has been studied. Each studied beach with a contribution of 150 tons of $CO_2$ per year becomes an important source of carbon into the 315 sea. Correspondingly, La Playa del Faro it is emitting 28 kg $CO_2$ $d^{-1}$ in each tidal flow to the atmosphere. It may seem very scarce compared to volcanic eruptions such as the most recent in the Canaries that occurred in the neighbouring island, El Hierro, in 2010, which was emitting $6.0 \times 10^5 \pm 1.1 \times 10^5$ kg $d^{-1}$ and now the emissions are unappreciated (Santana-Casiano et al., 2016). However, the flux of $CO_2$ from La Palma island seems to have been started before the islands were conquered in 1493 (Soler, 2007), being in a more advanced degassing phase than El Hierro, with fewer emissions, but continued over time. 320 Therefore, if we consider its timescale, La Palma becomes a significant $CO_2$ source. For all these reasons, PFS and the lagoons are an interesting area for future hydrological and oceanographic research, helping in new studies focusing on groundwater fluxes, the oceanic water cycle and oceanic carbon fluctuation (Moore, 2010; Santana-Casiano et al., 2016; Mulligan et al., 2019).

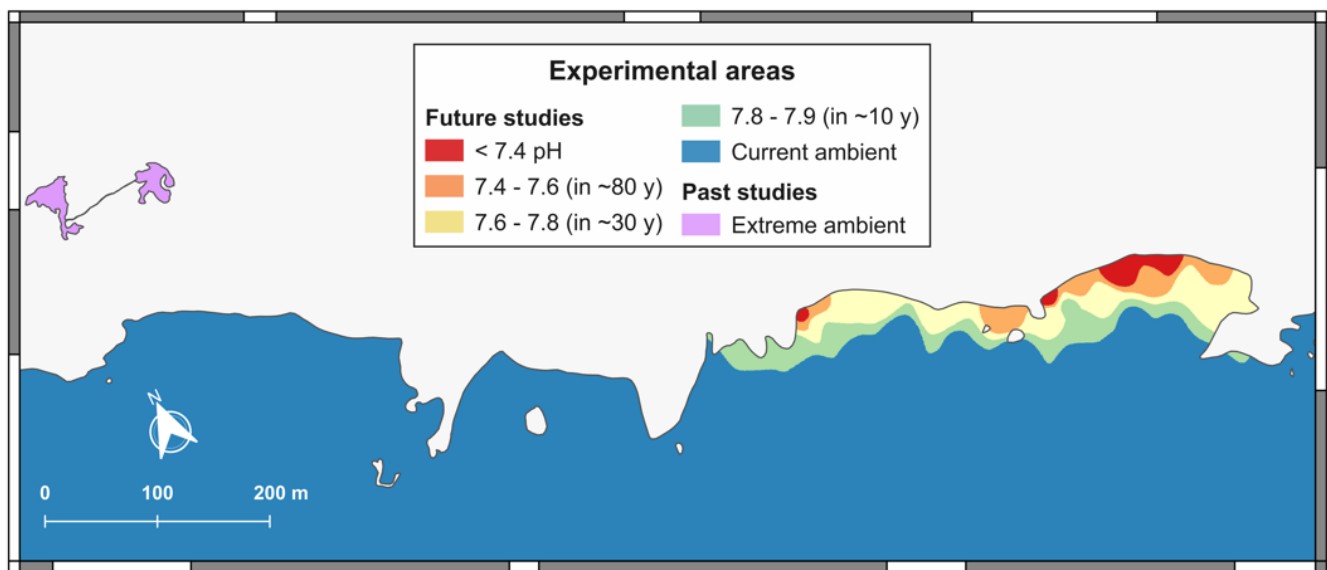

**Figure 6: Selected areas for experimental purpose (Interpolation IDW, 4.0 of correlation with Qgis).**

**5 Conclusions**

The studies carried out show the existence of continuous natural acidification in the southern coast of La Palma. This acidification process is caused by two natural phenomena: the discharge of submarine brackish waters from the aquifer and the magmatic emissions of $CO_2$ gas. Therefore, the monitoring of both sources is important not only from the biological point 330 of view but also from an atmospheric, oceanographic, volcanologist and hydrological perspective. The groundwater discharges found in Playa del Faro and Los Porretos (PFS) have similar chemical properties (even when alkalinity does not remain constant) that create a natural pH gradient analogous to future oceans conditions. Consequently, they can be used as natural laboratories to predict the effects of OA on the functioning of future oceans. In addition, the interior Echentive lagoons where

the chemical alterations are intensified, present the conditions capable of disentangling how life has persisted during higher
atmospheric $CO_2$ periods on planet Earth.

## Appendix A

**Table A1: Summary of the sampling methodology, with the locations sampled ("Sites"), the date of each sampling ("Date"), whether the sampling was done during the low (LT) or high (HT) tide and whether the parameters measured ("Measures") were all (ALL) or only the pH (pH).**

| Sites | Date | Tide | Nº | Measures |
|---|---|---|---|---|
| **Playa del Faro** | mar-18 | LT | 23 | All |
| **Playa del Faro** | dec-18 | HT | 17 | All |
| **Playa del Faro** | dec-18 | LT | 19 | All |
| **Playa del Faro** | jun-19 | HT | 11 | pH |
| **Playa del Faro** | jun-19 | LT | 11 | pH |
| **Los Porretos** | mar-18 | LT | 5 | All |
| **Los Porretos** | dec-18 | LT | 14 | All |
| **Los Porretos** | dec-18 | HT | 10 | All |
| **Echentive lagoon 1** | mar-18 | LT | 8 | All |
| **Echentive lagoon 1** | dec-18 | LT | 10 | All |
| **Echentive lagoon 1** | dec-18 | HT | 10 | All |
| **Echentive lagoon 1** | jun-19 | LT | 6 | pH |
| **Echentive lagoon 2** | dec-18 | LT | 6 | pH |
| **Echentive lagoon 2** | dec-18 | HT | 6 | pH |


## Appendix B

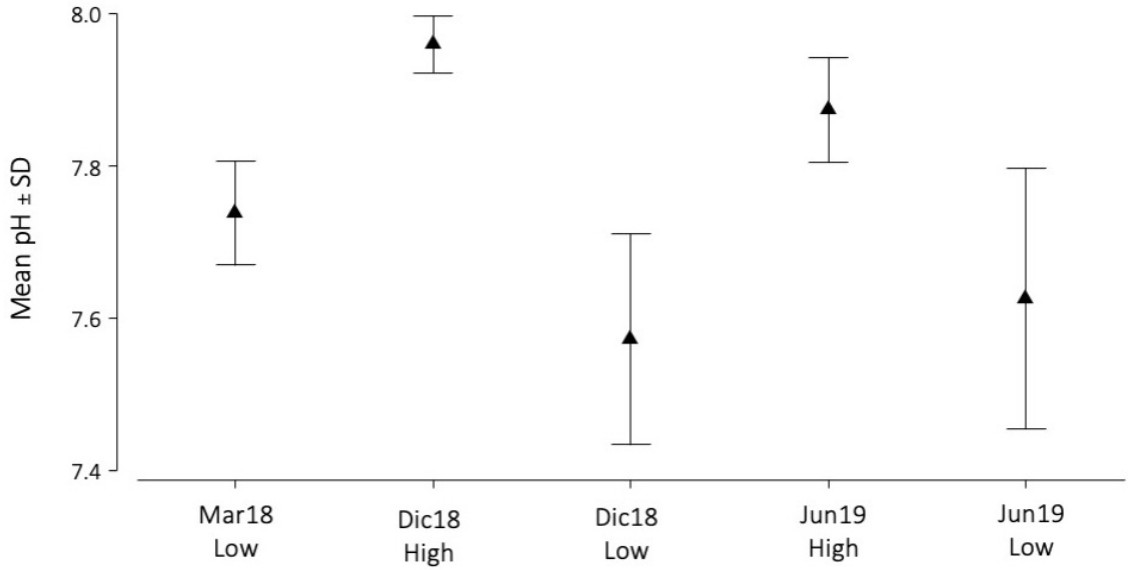

**Figure B1: Graph representing the tidal fluctuation (Low and High tide) of the mean pH with standard deviation (SD) at Playa del Faro, during March 2018 (Mar18), December 2018 (Dec18) and June 2019 (Jun19).**

**Data availability**

All measures obtained and used in this work will be published as supplementary material.

**Author contribution**

Sampling and data analysis were performed by all authors. SGD and JCH lead the paper writing and all authors contributed to the interpretation of the results and writing.

**Competing interests**

The authors declare that they have no conflict of interest.

**Acknowledgements**

This research received a grant from the Fundación Biodiversidad of the Ministerio para la Transición Ecológica of the Spanish Government and help from the Ministerio de Economía y Competitividad through ATOPFe project (CTM2017-83476). In March 2018, we thank to the officers, crew and researcher of the R/V Ángeles Alvariño from Instituto Español de Oceanografía (IEO) for their help during sampling process, specially Dr. E. Fraile and F. Domingo (from VULCANA-II-0318 project). Also, we want to thank Adrián Castro for his help during the water sample analysis in the laboratory of QUIMA group (ULPGC)
and Dr. E. Lozano-Bilbao from the University of La Laguna for his comments and feedback. Finally, we very much appreciate all the help offered by the Fuencaliente town hall (La Palma).

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
