# Peer review of "Chemical characterization of Punta de Fuencaliente CO2 seeps system (La Palma Island, NE Atlantic Ocean): a new natural laboratory for ocean acidification studies"

_Biogeosciences, 2020_

## Referee Comment (RC1) · Sara González-Delgado et al. · 24 Aug 2020

1. GENERAL COMMENTS (overall quality)

This study provides a novel and comprehensive description of a location resembling fu-
ture water chemistry conditions, as expected under ocean acidification scenarios. The
authors provide a valuable dataset of measured and estimated parameters in seven
sites, along the south of La Palma island, located in the North Atlantic Ocean. It is very
interesting the explanation the authors provided about the origin/source of these acidified waters, also the discussion of the community assemblages inside the lagoons, where conditions behave different from coastal waters.

2. SPECIFIC COMMENTS (individual scientific questions/issues)

Abstract: in the same way the authors presented the CO2 emission flux range in this section, it is advised to include the general range of measured and calculated parameters. This provides the reader with a general overview of the chemistry conditions in this location. - Please specify whether omega aragonite and calcite were measured or calculated.

Keywords: consider removing the word "area". Also, including the word "groundwater" to the list.

Material and methods:

- Was the VINDTA a 3C? If yes, please specify. - Authors are advice to include further details regarding water sampling and handling: sampling procedure (Niskin, SCUBA, etc), sampling containers for AT, CT and salinity (type of bottles), total number of samples (N per site, period, etc, consider present a summary of this information in a table as Supplementary material), samples fixed with HgCl2?, storing conditions. - There is no mentioned in this section of how they obtained the atmospheric CO2 values. This should be clarified.

Results:

- The authors indicated they found important differences between tides. This is an important finding, in agreement with results previously reported by Manzello (2010) in a shallow tropical coral reef, therefore, the authors could include an additional graphic representation of it as supplementary material (box plot, scatter plot or other). - The authors indicated that "Los Potreros, is a continuation of Playa del Faro", however, according to Fig.1 Los Barqueros is located between these locations.

Figures:

- It's unclear, what it is the purpose of the dashed-line square in the figures? In Figs.2, 4, 5 is used as division for different sites but in Fig. 3 represents a tide difference. The authors should try to standardize the use of this element among all figures and also be clearly indicated in the caption. - Fig.1: caption must include a description of the figures in each panel. - Fig.1: in order to facilitate reader's interpretation, ID letters from panel b and c should coincide. Currently, there is no clear whether the authors tried to make these panels complement of each other. For example, when interpreting the left map from panel c based in color/letter code (using yellow mark as reference to Playa Echentive), it seems there is a mixed up (the stars should be Lagoon1 and Lagoon 2, but currently are marked as Playa del Faro + Lagoon 2). Authors should carefully review the ID letters/colors from panels b and c. Another suggestion it's to merge both legends, by including the color code next to the letter in the legend from panel b. - Fig.1, Fig.5: it's unclear to which sampling period corresponds the panel "High tide". The authors should consider including tide initials (LT, HT) in all the panels/figures, maybe next to the sampling period title, and indicate it in the caption description. - Fig4: low and high tide labels are missing in the figure panels. - I would rather to see the order of the figures arranged by parameters. For example, move up Fig.5 after Fig.2, so all pH figures are shown together. This would facilitate following the figures, specially considering that arrangement per site does not follow the same order in all figures (Fig. 2 = Playa del Faro + Los Potreros but Fig. 3 = Los Potreros + Echentive Laggon 1, etc). - Fig. 8: caption requires minor modifications. "Selected" instead of "Select" and "Purpose" or "proposal" instead of "purpose".

3. TECHNICAL CORRECTIONS (typing errors, etc.)

- General: values <10 must be written in letters. - Line 20: start new sentence with "This". - Line 30: move "Since the last decade" to the beginning of the paragraph. Otherwise, it seems that you are referring to the effects exclusively taking place during the last decade. - Paragraph 50: replace "are" by a "," and move "are" in front of "an oceanic". - Paragraph 55: add "," before "which". - Paragraph 70: last sentence, add

"," before "what", and close the sentence with "?". - Paragraph 80: "were" instead of "where". - Study area: figures within the text are not mentioned in sequential order (1c comes prior 1b). Authors must either a) modify the order of the sentences in the text or b) exchange the panels order in the figure (swap 1b by 1c). - Line 100: "culometric" is missing an "o" after the "c" (typo). Tittle of Dickson's manual is incorrect. - Paragraph 105: remove "with" after "data using". - Paragraph 115: "during the last eruption" instead of "of the last eruption". - Paragraph 130, 240: use the same amount of decimal positions when reporting values (pH 8.0, omega calcite 5.0). - Paragraph 160: add "up" before "to". - Paragraph 165: "data only from" instead of "data from only". - Paragraph 190: add "," after "therefore". Remove "was" after "water"? - Paragraph 230: replace "a" by "an" before "unique". - Paragraph 240: remove "the" before "shore". - Paragraph 285: remove "s" in "predicts".

―――――――――――――――――――――――

---

## Referee Comment (RC2) · Anonymous Referee #2 · 30 Aug 2020

The Authors performed some measurements of the seawater carbonate chemistry around Punta de Fuencalente CO2 seeps. It focused on the role of groundwater discharge in the acidification of the local beaches. The aim was to describe a new natural analogue to study future (and past) conditions. While the description of such a system is welcome as each extreme site could add insight toward a better understanding of the mechanisms involved in the stress response, this system is far to be a natural analogue to study the effect of OA. With this in mind, I suggest the Authors to revise the ms

according to what they recognised to be central in their study (L 273), and improve the methods description, figure legends, which make this ms hard to be follow. However, I think the data are good and merit to be published. Some suggestions in a puzzling order: I do not understand the sentence in L 20. Both CO2 seeps and acid brackish water contribute to change the seawater chemistry! Wow! Authors should be more cautious about certain ideas. It is hard to think that these kind of systems could be used to understand how life persisted through past Eras. Ok for potential future scenarios, but with several assumptions.. L 31 and 34. The best references for this general sentence are Hall-Spencer et al 2008 and Dando et al 1999 respectively. Note for the Authors. It would be great to see here the relevant literature instead of Gonzales-Delgado and Hernandez 2018 only. For instance, Vizzini et al 2016, Pichler et al 2019 should be cited with regard to the potential biases of trace elements at seeps. L 64-68. This part is not clear. It suggests that the lagoons receive fresh inland ground water and a slight dilution of the seawater in the lagoons, so it receive water from both. Then the author state that "this indicate that the system is affected by the submarine groundwater, which probably originate from the thermal waters". What exactly the Authors want to say? And, without the data found (we are in the introduction) is it quite speculative, isn't it? L 65. "there are brackish lagoon located .. about 22m from the coastline", actually within 50 m in Fig. 1, and 100 m in L 229. Methods. This section needs to be deeply improved. The sampling methods and analyses need to be described. Fig. 1. I understand that panel left in c represents the lagoon, but what about panel right? The legend should contain more details and the figure should be self-explained. What is the role of the two identical stars in panel right which are repeated in fig 2 in all the sites? Are the figures with colour? It is difficult to read the pH etc. Nice work putting lat & long but meters would be better to directly appreciate the extension of the area. Authors wrote that sampling were performed between 0 and 2 m depth (need details in the methods). Is the sites so shallow everywhere? Considering the 2 m oscillation in the tide, why the Authors only sampled the intertidal zone? Fig. 1 vs results. Well it is hard. Ok, sites Playa del Faro, Los Porretos, Lagoon 1 and 2 (also Enchentive, called

Playa Echentive in Fig 1 table b); Las Cabras?? Last eruption was in the 17th century? L 116. "In all cases, the anomalies were the highest during low tide." Please change the word anomalies. So what? Where are the seeps? On the beach? Their extensions? Their depth? Salinity was 31 in the lagoons and normal near the coast. These measurements did not suggest any link between lagoon and the beach. So, L 127-128 how the Authors can state that the SW carbonate chemistry was strongly affected by the entrance of water with less salinity? L 141. "During high tide, the anomalies almost disappeared.." which support the hypothesis about the role of the lagoon in the local beach acidification. But, if they exist, what about the CO2 seeps it was supposed to acidify the area? Fig. 7 the figure is fairly useless and does not describe the role of the lagoons. L. 202. PFS. Please just write the location. L. 205. I agree with the fact that this system is similar to the Ojos in Mexico. The latter has been a highly debated "natural analogue" to future conditions since the groundwater discharge profoundly change the seawater chemistry and do not mimic what we should expect in the future. CO2 seeps are more "realistic" in some ways and with limitations. I invite the Authors to pay attention about this potential caveat when using the PFS as a natural lab to study the effect of ocean acidification. Paragraph 4.3. Sorry but La Palma is not similar to other natural acidified systems, and I do not believe it is a very useful spot for large-scale and long-term adaptation experiments...to be used as an analogue of climate change scenarios. Please, be objectives. For instance, although the data are nice and I understand the effort put in such a sampling, from this data set it is not clear what is the real variability in time and space (L 238: PFS have been characterized from the shore to offshore.. is not really true, at least from what I understood by reading the few details given in the methods). The Authors suggested some of these caveats in the 20 lines from L244, which is good. In the discussion (paragraph 4.2) some speculative observations about the community are described. It is complicate to appreciate the site as a natural lab with only such a scarce description of the biota. Then, L 259 the Authors added this sentence: "only one type of rocky benthic habitat is present.." Well, we know that OA will affect the marine organisms (maybe) but I think this is too much! Maybe

there are some caveats in using this interesting site as a natural analogue. The last sentence (L 273) is, in my opinion the best one describing the aim of this study. I invite the Authors to revise the paper in the direction they finally described. Conclusions. I disagree with most of its content.

---

## Referee Comment (RC3) · Sylvain Agostini (Referee) · 9 Sep 2020

The authors describe the chemistry of a "new" CO2 vent system. Due to the extreme variability at all sites and the change in alkalinity, the relevance of these sites as a laboratory for future ocean acidification seems limited. Most of the locations seems to have already been described in previous publications. Perhaps the only new location is the lagoon site but its use as a natural analogue for past and future oceans is questionable due to the addition of brackish and groundwater. The other locations were already reported, following the nomenclature in Figure 1:

* site H is reported in Hernández, C. A., C. Sangil, and J. C. Hernández. 'A New CO2 Vent for the Study of Ocean Acidification in the Atlantic'. Marine Pollution Bulletin 109, no. 1 (15 August 2016): 419–26. https://doi.org/10.1016/j.marpolbul.2016.05.040.

* Site A, B are reported in Viotti, Sofía, Carlos Sangil, Celso Agustín Hernández, and José Carlos Hernández. 'Effects of Long-Term Exposure to Reduced PH Conditions on the Shell and Survival of an Intertidal Gastropod'. Marine Environmental Research 152 (1 December 2019): 104789. https://doi.org/10.1016/j.marenvres.2019.104789.

* E,F,G data are not reported in this manuscript as far as I can see.

As it is not evident what data is novel, please clearly state what part of the data is unpublished and novel data and which one is not. Also please clearly highligt what does the additional chemistry data add to the previously published studies. At the moment I have difficulties in recommending this manuscript for publication.

---

## Author Comment (AC1) · 1 Oct 2020

Thank you for your revision of our manuscript. Your comments have been essential in guiding our revision and we hope that we have satisfactorily dealt with the errors and clarify any point that was originally confusing. We very much appreciate your effort in refining our research and to improve our text. We also thanks that you have found the subject of the review manuscript interesting, as you state: "This study provides a novel

and comprehensive description of a location resembling future water chemistry conditions, as expected under ocean acidification scenarios. The authors provide a valuable dataset of measured and estimated parameters in seven sites, along the south of La Palma island, located in the North Atlantic Ocean. It is very interesting the explanation the authors provided about the origin/source of these acidified waters, also the discussion of the community assemblages inside the lagoons, where conditions behave different from coastal waters."

Responses to Referee's comments:

SPECIFIC COMMENTS (individual scientific questions/issues)

Referee #1; comment 1: "Abstract: in the same way the authors presented the CO2 emission flux range in this section, it is advised to include the general range of measured and calculated parameters. This provides the reader with a general overview of the chemistry conditions in this location. - Please specify whether omega aragonite and calcite were measured or calculated."

- Response: We agree with the comment and we have made the changes. Please see sentences between Line 18 and 21.

Referee #1; comment 2: "Keywords: consider removing the word "area". Also, including the word "groundwater" to the list."

- Response: We agree with you and we have made the changes. Check Line 28.

Referee #1; comment 3: "Material and methods:- Was the VINDTA a 3C? If yes, please specify. - Authors are advice to include further details regarding water sampling and handling: sampling procedure (Niskin, SCUBA, etc), sampling containers for AT, CT and salinity (type of bottles), total number of samples (N per site, period, etc, consider present a summary of this information in a table as Supplementary material), samples fixed with HgCl2?, storing conditions. - There is no mentioned in this section of how they obtained the atmospheric CO2 values. This should be clarified."

- Response: We agree with your comments and we have added more details about the sampling and handling methods. Please go to Lines 105 - 109 and Lines 113 -115, 119 and Lines 123 - 124. Also, we now include a table with your suggestion (please see supplementary material 2 - Table SM2).

Referee #1; comment 4: "Results: - The authors indicated they found important differences between tides. This is an important finding, in agreement with results previously reported by Manzello (2010) in a shallow tropical coral reef, therefore, the authors could include an additional graphic representation of it as supplementary material (box plot, scatter plot or other). – The authors indicated that "Los Potreros, is a continuation of Playa del Faro", however, according to Fig.1 Los Barqueros is located between these locations."

- Response: We agree with your suggestion and we have included a line graph with the tidal fluctuation of the minimum pH (see supplementary material 2 – Figure SM2). We apologize for the confusion between "Los Porretos" and "Los Barqueros" both are the same site. We have made the correction in Figure 1.

Referee #1; comment 5: " Figures: - It's unclear, what it is the purpose of the dashed-line square in the figures? In Figs.2, 4, 5 is used as division for different sites but in Fig. 3 represents a tide difference. The authors should try to standardize the use of this element among all figures and also be clearly indicated in the caption. - Fig.1: caption must include a description of the figures in each panel. - Fig.1: in order to facilitate reader's interpretation, ID letters from panel b and c should coincide. Currently, there is no clear whether the authors tried to make these panels complement of each other. For example, when interpreting the left map from panel c based in color/letter code (using yellow mark as reference to Playa Echentive), it seems there is a mixed up (the stars should be Lagoon1 and Lagoon 2, but currently are marked as Playa del Faro + Lagoon 2). Authors should carefully review the ID letters/colors from panels b and c. Another suggestion it's to merge both legends, by including the color code next to the letter in the legend from panel b. - Fig.1, Fig.5: it's unclear to which sampling period

corresponds the panel "High tide". The authors should consider including tide initials (LT, HT) in all the panels/figures, maybe next to the sampling period title, and indicate it in the caption description. - Fig4: low and high tide labels are missing in the figure panels. - I would rather to see the order of the figures arranged by parameters. For example, move up Fig.5 after Fig.2, so all pH figures are shown together. This would facilitate following the figures, specially considering that arrangement per site does not follow the same order in all figures (Fig. 2 = Playa del Faro + Los Potreros but Fig. 3 = Los Potreros + Echentive Laggon 1, etc). - Fig. 8: caption requires minor modifications. "Selected" instead of "Select" and "Purpose" or "proposal" instead of "purpose".

- Response: We agree with your comment and we hope that we have satisfactorily clarify this confusion. Figure 1 has been modified; the identification letters now match between panel a and b, and high and low tide are marked as HT and LT respectively. Figures 2, 3 and 4 have been modified in line with your comments: now Figure 2 has all pH interpolation graph. In Figures 3 and 4 we have added the initials LT and HT in the titles of each graph. Also, the dashed line has only one purpose, to separate each sampling site. Finally, we have corrected the caption of Figure 8 in Line 293.

TYPING ERRORS, ETC.

Referee #1; comment 6: "General: values <10 must be written in letters. - Line 20: start new sentence with "This". - Line 30: move "Since the last decade" to the beginning of the paragraph. Otherwise, it seems that you are referring to the effects exclusively taking place during the last decade. - Paragraph 50: replace "are" by a "," and move "are" in front of "an oceanic". - Paragraph 55: add "," before "which". - Paragraph 70: last sentence, add "," before "what", and close the sentence with "?". - Paragraph 80: "were" instead of "where". - Study area: figures within the text are not mentioned in sequential order (1c comes prior 1b). Authors must either a) modify the order of the sentences in the text or b) exchange the panels order in the figure (swap 1b by 1c). - Line 100: "culometric" is missing an "o" after the "c" (typo). Tittle of Dickson's manual is incorrect. – Paragraph 105: remove "with" after "data using". - Paragraph 115: "during

the last eruption" instead of "of the last eruption". - Paragraph 130, 240: use the same amount of decimal positions when reporting values (pH 8.0, omega calcite 5.0). - Paragraph 160: add "up" before "to". - Paragraph 165: "data only from" instead of "data from only". – Paragraph 190: add "," after "therefore". Remove "was" after "water"? - Paragraph 230: replace "a" by "an" before "unique". - Paragraph 240: remove "the" before "shore". – Paragraph 285: remove "s" in "predicts".

- Response: We thank you very much for providing these text corrections. We have corrected the errors in Lines 22, 31, 51, 58, 77 - 78, 83, 92 – 96, 113, 115, 117, 121, 130, 136 – 253, 171, 174, 204, 205, 244 - 245, 252, 300.

Please also note the supplement to this comment:
https://bg.copernicus.org/preprints/bg-2020-232/bg-2020-232-AC1-supplement.pdf

―――――――――――――――

[Figure]

Fig. 1.

[Figure]

**Fig. 2.**

[Figure]

**Fig. 3.**

[Figure]

Fig. 4.

**Supplement:**

**Table SM2: Summary of the sampling methodology, with the locations sampled ("Sites"), the date of each sampling ("Date"), whether the sampling was done during the low (LT) or high (HT) tide and whether the parameters measured ("Measures") were all (ALL) or only the pH (pH).**

| Sites | Date | Tide | Nº | Measures |
|---|---|---|---|---|
| **Playa del Faro** | mar-18 | LT | 23 | All |
| **Playa del Faro** | dec-18 | HT | 17 | All |
| **Playa del Faro** | dec-18 | LT | 19 | All |
| **Playa del Faro** | jun-19 | HT | 11 | pH |
| **Playa del Faro** | jun-19 | LT | 11 | pH |
| **Los Porretos** | mar-18 | LT | 5 | All |
| **Los Porretos** | dec-18 | LT | 14 | All |
| **Los Porretos** | dec-18 | HT | 10 | All |
| **Echentive lagoon 1** | mar-18 | LT | 8 | All |
| **Echentive lagoon 1** | dec-18 | LT | 10 | All |
| **Echentive lagoon 1** | dec-18 | HT | 10 | All |
| **Echentive lagoon 1** | jun-19 | LT | 6 | pH |
| **Echentive lagoon 2** | dec-18 | LT | 6 | pH |
| **Echentive lagoon 2** | dec-18 | HT | 6 | pH |

[Figure]

**Figure SM2: Line graph representing the tidal fluctuation of the minimum pH values (Min pH) at Playa del Faro, during March 2018 (Mar-18), December 2018 (Dec-18) and June 2019 (Jun-19).**

---

## Author Comment (AC2) · 1 Oct 2020

Thank you for reviewing our manuscript, we really appreciate your effort in refining our research. Your comments have been very constructive and help us to improve our research. We hope that we have satisfactorily dealt with the original confusing points. We also thanks that you have found our data worth publishing, even though there was a misunderstanding, caused by some typing errors and confusion in the figures.

[Figure]

Responses to Referee's comments:

Referee #2; comment 1: "The Authors performed some measurements of the seawater carbonate chemistry around Punta de Fuencalente $CO_2$ seeps. It focused on the role of groundwater discharge in the acidification of the local beaches. The aim was to describe a new natural analogue to study future (and past) conditions. While the description of such a system is welcome as each extreme site could add insight toward a better understanding of the mechanisms involved in the stress response, this system is far to be a natural analogue to study the effect of OA. With this in mind, I suggest the Authors to revise the ms according to what they recognised to be central in their study (L 273), and improve the methods description, figure legends, which make this ms hard to be follow. However, I think the data are good and merit to be published."

- Response: We disagree with "this system is far from being a natural analogue for studying the effect of OA", as we have shown in this manuscript, this Fuencaliente area is acidified due to the volcanic activity which is altering the groundwater that is continuously being discharged into the shore. Please note that being "analogue" is not the same as being "equal", we consider La Palma system an "analogue" and similar to other natural analogues, such as Ischia seeps system, Papua New Guinea $CO_2$ vents or Puerto Morelos acidify system. All of them are special places because they present pH and $pCO_2$ values similar to future IPCC predictions, as we have demonstrated for La Palma seeps system, and despite the anomalies that all of these natural systems present (see Table 2 from González-Delgado and Hernández, 2018). On the other hand, regarding the description of the methods as well as the legend of the figure, we have followed your recommendations and those of the Referee #1 and have made the appropriated changes to better explain these sections.

Referee #2; comment 2: " I do not understand the sentence in L 20. Both $CO_2$ seeps and acid brackish water contribute to change the seawater chemistry! Wow! Authors should be more cautious about certain ideas."

- Response: We agree with you and we have changed the sentence for better understanding (see Line 22 - 23). In this manuscript, we have demonstrated that volcanic $CO_2$ emissions alter the brackish groundwater that is discharged into the coast of La Palma, changing the chemistry of the water (see Figure 5 for understand the process and Figure 2 to 4 for the data).

Referee #2; comment 3: "It is hard to think that these kind of systems could be used to understand how life persisted through past Eras. Ok for potential future scenarios, but with several assumptions.."

- Response: We disagree with the comment in general. As we have already mentioned in the manuscript, within La Palma system we have found a very extreme environment in the Echentive lagoons. These extreme chemical characteristics (for example, Ct values of 10817.12 $\mu$mol kg-1 and pH of 7.12 unit) could be used to understand how life has persisted in these extreme conditions, similar to Rio Tinto in Spain or the hot springs in Yellowstone. The study of the extremophiles organisms that live there can help us to understand how was the beginning of life on earth. We believe this is an interesting research topic and worth to mention in the manuscript.

Referee #2; comment 4: "L 31 and 34. The best references for this general sentence are Hall-Spencer et al 2008 and Dando et al 1999 respectively. Note for the Authors. It would be great to see here the relevant literature instead of Gonzales-Delgado and Hernandez 2018 only. For instance, Vizzini et al 2016, Pichler et al 2019 should be cited with regard to the potential biases of trace elements at seeps."

- Response: We agree with your suggestion and we have included more relevant literature. Please go to Lines 32-33, 36, 39, 47-48.

Referee #2; comment 5: "L 64-68. This part is not clear. It suggests that the lagoons receive fresh inland ground water and a slight dilution of the seawater in the lagoons, so it receive water from both. Then the author state that "this indicate that the system is affected by the submarine groundwater, which probably originate from the thermal

waters". What exactly the Authors want to say? And, without the data found (we are in the introduction) is it quite speculative, isn't it?"

- Response: We agree that it is possible to clarify this part. In this section, we try to explain that there is a mixture of seawater with brackish groundwater in the Echentive Lagoons and there are previous evidences of this finding since Soler (2007) and Calvet et al., (2003) studies. We apologize for the confusion and we have made the changes to Line 69 - 70.

Referee #2; comment 6: "L 65. "there are brackish lagoon located .. about 22m from the coastline", actually within 50 m in Fig. 1, and 100 m in L 229."

- Response: We agree that there was a problem in figure 1 and in the text and we have changed both (see Figure 1 and Line 243). Line 68 is the correct one "...at about 200 m from...".

Referee #2; comment 7: "Methods. This section needs to be deeply improved. The sampling methods and analyses need to be described."

- Response: We agree with your comments and we have added more details about sampling and handling methods following your comment and those of Referee #1. Please go to Line 92 - 124.

Referee #2; comment 8: "Fig. 1. I understand that panel left in c represents the lagoon, but what about panel right? The legend should contain more details and the figure should be self-explained. What is the role of the two identical stars in panel right which are repeated in fig 2 in all the sites? Are the figures with colour? It is difficult to read the pH etc. Nice work putting lat & long but meters would be better to directly appreciate the extension of the area."

- Response: We agree that the caption figures needed more details and we made the modifications accordingly (see Line 97 - 102 and Line 184 - 193). We believe that now the figures are self-explanatory. The stars was included to better interpret and locate

the interpolation graphics from Figure 2 - 4. All the figures have vivid colors for a better interpretation of the elements. For the interpolations graphs, it is possible to see the anomaly using a color gradient from red to blue. We have corrected some errors in the legend and scales of some panels in Figure 1. Now, you can see the extension of the affected area in meters (see Figure 1).

Referee #2; comment 9: "Authors wrote that sampling were performed between 0 and 2 m depth (need details in the methods). Is the sites so shallow everywhere? Considering the 2 m oscillation in the tide, why the Authors only sampled the intertidal zone?"

- Response: We agree with your comments and we have further explained the sampling process (please see Line 104 - 108). In figure 1 you can see that we took samples from the shore up to 50 m inland (and one control point up to 200 m). Scuba dive was used to take the water samples with the bottles between 0 and 2 m depth. This samples were taken in the beach, no at the intertidal zone.

Referee #2; comment 10: "Fig. 1 vs results. Well it is hard. Ok, sites Playa del Faro, Los Porretos, Lagoon 1 and 2 (also Enchentive, called Playa Echentive in Fig 1 table b); Las Cabras?? Last eruption was in the 17th century? L 116. "In all cases, the anomalies were the highest during low tide." Please change the word anomalies. So what? Where are the seeps? On the beach? Their extensions? Their depth? Salinity was 31 in the lagoons and normal near the coast. These measurements did not suggest any link between lagoon and the beach. So, L 127-128 how the Authors can state that the SW carbonate chemistry was strongly affected by the entrance of water with less salinity?"

- Response: Previous figures have been improved to avoid misunderstanding regarding the location of the samples and the anomalies. The errors you highlighted have been corrected (please see Figures 1, 2, 3 and 4). Playa Echentive and Echentive lagoons are two different sampling sites and we have corrected it on the figures and on the text.
Las Cabras, as explained in the manuscript in Lines 59 - 60, is a CO2 seep recently described by Hernández et al., 2016, and it was sampled again for us (see Line 92 - 104 and Figure 1). However, as we already explained in the manuscript, "Las Cabras site was discarded in subsequent samplings due to the difficult access, the poor sea conditions and the small size of the area affected by the emissions (Hernández et al. 2016)". Furthermore, as it explained in the manuscript "These four areas, Las Cabras, La Playa del Faro, Los Porretos and the two Echentive Lagoons (Fig. 1b,c), correspond to areas that were not buried by the lava during the last eruption (Teneguía volcano 1971; Padrón et al., 2015, Fig. 1b)", so the last eruption was in the 20th century.

We do not understand why we might have to change the word "anomalies". Anomaly means " a...thing that is different from what is usual..." (Cambridge Dictionary). So, we think that it is a good word to use when the salinity, pH, pCO2, CT, AT, $\Omega$calcite and $\Omega$aragonite exceed the normal values for seawater; as it happens in the seeps found in the beaches of Las Cabras, La Playa del Faro, Los Porretos and the two Echentive Lagoons (please see Figure 2, 3 and 4 and supplementary material 3). With regard to salinity, the lowest values found in the Echentive lagoon are 31 - 32 units of salinity. However, it has also been detected, as it been said in the manuscript "... slightly less saline water near the coast." with value of 36.51 – 36. 07 during low tide (see supplementary material 3). It can be thought that this is a normal value of salinity, nonetheless we see that, during the high tide and in the control areas, the salinity is always higher (37.05 units) (please see supplementary material 3). For this reason, we disagree with the last sentence of your comment and we want to remark that "...the entrance of water with less salinity" with very extreme values of pH, pCO2, CT, AT, $\Omega$calcite and $\Omega$aragonite near the coast, especially during low tide in Playa del Faro and Los Porretos exist and strongly change the chemistry of seawater. All our results demonstrated this fact (see Figure 2, 3, 4 and supplementary material 2 and 3).

Referee #2; comment 11: "L 141. "During high tide, the anomalies almost disappeared.." which support the hypothesis about the role of the lagoon in the local beach

acidification. But, if they exist, what about the CO2 seeps it was supposed to acidify the area? Fig. 7 the figure is fairly useless and does not describe the role of the lagoons."

- Response: Again, we believe there is a misunderstanding. At no point we have suggested that the lagoons play a role in the acidification of beaches. Our hypothesis would be, as it been said in the manuscript, "….what occurs in areas where SGD is enriched by the emissions of recent volcanism or by hydrothermal activity?... these discharges can also act as sources of gases and hydrothermal emission compounds to the ocean and become points of emission of CO2 that contribute to the OA". Therefore, what we have defended in this paper is that a source of brackish groundwater that is affected by volcanic emissions, seeps through the soil and rocks into the sea (see Figure 5). On the way, it accumulates forming the Echentive lagoons where it mixes with seawater. The old Figure 7, now Figure 5 shows us a drawing of the process of acidification of the beaches. It seems clear then that CO2 gases, from volcanic activity, are mixing with brackish groundwater that are discharged o seeped in the coast through the rock porosity.

Referee #2; comment 12: "L. 202. PFS. Please just write the location."

- Response: We agree with the comment and made the change in Line 216.

Referee #2; comment 13: "L. 205. I agree with the fact that this system is similar to the Ojos in Mexico. The latter has been a highly debated "natural analogue" to future conditions since the groundwater discharge profoundly change the seawater chemistry and do not mimic what we should expect in the future. CO2 seeps are more "realistic" in some ways and with limitations. I invite the Authors to pay attention about this potential caveat when using the PFS as a natural lab to study the effect of ocean acidification."

- Response: We agree that CO2 seep can be more "realistic" than the Ojos system in Mexico when studying the effect of ocean acidification. Nevertheless, as we have emphasized in the manuscript and throughout the responses to your comments, PFS can be considered a CO2 seep system, because the CO2 emissions that altered the

groundwater comes from volcanic activity. The PFS clearly mimic what we should expect in the future, as it has been demonstrated with our study. To improve this interpretation, we have made changes to Lines 220 and 226. We also wants to clarify, again, that we have pay attention to potential caveat and it have been highlighted in the manuscript (please go to Line 256 to 276). Therefore, and although not perfect (as the rest of the natural acidified systems already described), PFS is an analogue of future oceans and it can be used to understand the impact of OA on marine organism or ecosystem functioning.

Referee #2; comment 14: "Paragraph 4.3. Sorry but La Palma is not similar to other natural acidified systems, and I do not believe it is a very useful spot for large-scale long-term adaptation experiments: : :to be used as an analogue of climate change scenarios. Please, be objectives. For instance, although the data are nice and I understand the effort put in such a sampling, from this data set it is not clear what is the real variability in time and space (L 238: PFS have been characterized from the shore to offshore.. is not really true, at least from what I understood by reading the few details given in the methods). The Authors suggested some of these caveats in the 20 lines from L244, which is good. "

- Response: We think that you have misinterpreted our work, possibly because of some errors found in the previous version of the manuscript, that have led to several confusions when interpreting the results. The lack of some details in the methodology or the figures did not help either. We hope that now our clarifications may help you to have a better interpretation of our work.

Referee #2; comment 15: "In the discussion (paragraph 4.2) some speculative observations about the community are described. It is complicate to appreciate the site as a natural lab with only such a scarce description of the biota. Then, L 259 the Authors added this sentence: "only one type of rocky benthic habitat is present.." Well, we know that OA will affect the marine organisms (maybe) but I think this is too much! Maybe there are some caveats in using this interesting site as a natural analogue. The last

sentence (L 273) is, in my opinion the best one describing the aim of this study. I invite the Authors to revise the paper in the direction they finally described. Conclusions. I disagree with most of its content."

- Response: In section 4.2, we consider that we have not made any "speculative observations". It is true that there is little description of the biota, yet we consider this work to be purely about the chemical and physical characteristics of the area. We are in the process of publishing another manuscript with a detailed description of the flora and fauna from the PFS. Therefore, in the old Line 259 there was another misunderstanding, so that this will not happen again, we have made the corresponding changes and added the missing reference (see Line 273). When we say that "only a type of rocky benthic habitat is present", we refer in a general sense to the typical habitat found in the south of La Palma Island, not to the marine communities presents.

Please also note the supplement to this comment:
https://bg.copernicus.org/preprints/bg-2020-232/bg-2020-232-AC2-supplement.zip

—————————————————

[Figure]

(a)

(b)

**Last eruption**
- Teneguía lava
- Land gained by the eruption

| Id | Site | Time | Mesures | Tide |
|---|---|---|---|---|
| A | Playa del Faro (out) | D | ALL | LT |
| B | Playa del Faro | MDJ | ALL | LT/HT |
| C | Los Porretos | MD | ALL | LT/HT |
| D | Los Porretos (out) | D | ALL | LT |
| E | Playa Echentive | D | pH | LT |
| F | Echentive lagoon 1 | MDJ | ALL | LT/HT |
| G | Echentive lagoon 2 | D | pH | LT |
| H | Las Cabras | M | pH | LT |

M: March 2018, D: December 2018, J: June 2019
LT: Low Tide, HT: High Tide

Teneguía Volcano

(c)

**Network**
- A
- B
- C
- D
- E
- F
- G

Punta de Fuencaliente system (PFS)

Selected area where
a volume equal to
19700 m³

☆ Mark for understanding interpolation graphs (Figure 2-4)

**Fig. 1.**

[Figure]

Fig. 2.

[Figure]

**Fig. 3.**

[Figure]

**Fig. 4.**

[Figure]

**Fig. 5.**

---

## Author Comment (AC3) · 1 Oct 2020

Thank you for the review our manuscript and for your comments and constructive criticism. We have considered them and add more information to clarify the confusing points.

Responses to Referee's comments:

[Figure]

Referee #3; comment 1: "The authors describe the chemistry of a "new" CO2 vent system. Due to the extreme variability at all sites and the change in alkalinity, the relevance of these sites as a laboratory for future ocean acidification seems limited. Most of the locations seems to have already been described in previous publications. Perhaps the only new location is the lagoon site but its use as a natural analogue for past and future oceans is questionable due to the addition of brackish and groundwater. The other locations were already reported, following the nomenclature in Figure 1: * site H is reported in Hernández, C. A., C. Sangil, and J. C. Hernández. 'A New CO2 Vent for the Study of Ocean Acidification in the Atlantic'. Marine Pollution Bulletin 109, no. 1 (15 August 2016): 419–26. https://doi.org/10.1016/j.marpolbul.2016.05.040. * Site A, B are reported in Viotti, Sofía, Carlos Sangil, Celso Agustín Hernández, and José Carlos Hernández. 'Effects of Long-Term Exposure to Reduced PH Conditions on the Shell and Survival of an Intertidal Gastropod'. Marine Environmental Research 152 (1 December 2019): 104789. https://doi.org/10.1016/j.marenvres.2019.104789. * E,F,G data are not reported in this manuscript as far as I can see. As it is not evident what data is novel, please clearly state what part of the data is unpublished and novel data and which one is not. Also please clearly highligt what does the additional chemistry data add to the previously published studies. At the moment I have difficulties in recommending this manuscript for publication."

- Response: We agree with you that there is some caveats in these type of natural acidified system, however these cavities exists in all of the already described systems. We recommend you to see the Table 2 of our review paper (González-Delgado and Hernández 2018 - Advances in Marine Biology) where we do a comparison between natural acidified systems worldwide. Although, not perfect, these systems with their cavities are very useful to study the impact of OA on marine organisms and its capacity of adaptation, among other things. And, are by far more realistic than OA in vitro experiments. Therefore, we do not agree with you and these systems can be considered natural analogues of future oceans.
It is true that site H (Las Cabras), as well as, sites B and C (Playa del Faro and Los Por-retos) have been previously reported. However, this study is the first detailed chemical characterization of the whole area and include new seeps (Echentive Lagune and Los Porretos). For the present study, we include data that have not been used before and that have been collected on the long-term and in a larger scale. Additionally, for this study pH, pCO2, temperature, alkalinity and salinity have been measured accurately using proper apparatus (e.g. VINDTA 3C for alkalinity). Therefore, we consider to be a novelty: (1) The precise chemical description of this acidified system composed of several CO2 seep points and, as you said, the description of the Echentive Lagoons (F and G). (2) All the data presented, and its spatial and temporal variability. (3) And the description of the process of acidification of the coastal area of Fuencaliente (Origin of the seeps).

We would like to clarify, again, that all the measurements in this work (see supplementary material 3) are unpublished data. And we believe that we have not at anywhere in the text given any indication to the contrary. It is true that there are pH and pCO2 measurements at Las Cabras and La Playa del Faro in the previous two papers. We have included this information in Lines 60 and 94. However, these measurements were made at another time and with a different, less precise, methodology and only at the sampling points.

Please also note the supplement to this comment:
https://bg.copernicus.org/preprints/bg-2020-232/bg-2020-232-AC3-supplement.zip

---

## Referee Report (RR1)

[revised manuscript text omitted]


375

**Figure B1: Graph representing the tidal fluctuation (Low and High tide) of the mean pH with standard deviation (SD) at Playa del Faro, during March 2018 (Mar18), December 2018 (Dec18) and June 2019 (Jun19).**

**Data availability**

All measures obtained and used in this work will be published as supplementary material.

**Author contribution**

Sampling and data analysis were performed by all authors. SGD and JCH lead the paper writing and all authors contributed to the interpretation of the results and writing.

**Competing interests**

395   The authors declare that they have no conflict of interest.

**Acknowledgements**

This research received a grant from the Fundación Biodiversidad of the Ministerio para la Transición Ecológica of the Spanish Government and help from the Ministerio de Economía y Competitividad through ATOPFe project (CTM2017-83476). In March 2018, we thank to the officers, crew and researcher of the R/V Ángeles Alvariño from Instituto Español de Oceanografía

400   (IEO) for their help during sampling process, specially Dr. E. Fraile and F. Domingo (from VULCANA-II-0318 project). Also, we want to thank Adrián Castro for his help during the water sample analysis in the laboratory of QUIMA group (ULPGC) and Dr. E. Lozano-Bilbao from the University of La Laguna for his comments and feedback. Finally, we very much appreciate all the help offered by the Fuencaliente town hall (La Palma).

[revised manuscript text omitted]

---

## Author Response (AR2)

Biogeosciences Ref. No.: bg-2020-232

Chemical characterization of Punta de Fuencaliente  $CO_2$  seeps system (La Palma Island, NE Atlantic Ocean): a new natural laboratory for ocean acidification studies / Sara González-Delgado et al.

**Responses to Peter Landschützer' revision:**

Comments to the Author: "Dear authors, I have now studied your revised manuscript with track changes and went back to the referee comments and your response as well as my own recommendation. To recap: The majority of referees considered the natural analogy comparison problematic and urged the authors to substantially revise the manuscript in this respect. Likewise, a second major concern was raised related the novelty of the study as some results on the sites are already published. In response to these referee concerns I further explicitly asked in my decision letter for you to substantially revise (i.e. major revisions) the manuscript to clarify these points. From my reading of your response to the referee comments and the final revised paper (following my editorial comment) I dont believe you have taken the comment into serious considerations. This is also fairly easy visible by the minor changes in the track-changed version of the manuscript. This has further been recognised by the referee comments I received from referee#2 who has reviewed the revised manuscript. Unfortunately, there were some technical issues, hence referee#2 has sent the comments to me electronically and only used the online system for the recommendation. Please find the comments at the end of this letter. I dont think it was necessary to change your position regarding the natural analogy assumption, however, some concerns from round 1 have not been (fully) addressed. While I read e.g. that the authors added some points on the limitations (lines 252-262), I dont believe this qualifies as the discussion the referees were looking for. E.g. in his referee letter, referee#3 is concerned about the "extreme variability" at these sites and how this effects the natural analogy assumption (in particular, the referee called the the usefulness as natural laboratory limited). In the response and the revised manuscript, however, I barely see this addressed (with the exception of the link to the tidal influence). I was also hoping that the authors would use this occasions to not simply cite the studies that have previously explored these and other sites, but also clearly discuss the new findings in comparison to previous studies (also at different sites). The referees clearly seem to be concerned about the natural analogy assumption and they refer back to previous literature and studies that do show that such an analogy can be problematic (see also the comments from referee#2 on the revised manuscript below). Again, I dont expect the authors to change their position that the site is a useful natural laboratory, but I expect the authors, in response to the referee concerns, to provide clear and solid evidence and a substantial discussion also in comparison to previous studies. Based on my reading and the referee comments I decided that my position has not changed since round 1, i.e. the authors need to conduct major corrections to the manuscript. All referees welcome the description of the site and the system (although I would welcome not only citing previous studies, but also highlighting how the new data advance the knowledge in comparison to these previous studies). In particular - and I cant stress this enough - I want the authors to further address the referee concerns about the natural analogies, e.g. the variability or "noise" as mentioned by the referees (referee#3 in the first round and referee#2 below) and have a fair and substantial discussion on the limitations also in light of other studies conducted (also at different sites).

**Response**: First of all, we apologise for not fulfil yours's and reviewer 2 previous requirements. We have now further addressed the two previous big controversial points: (1) the usefulness of this kind of acidified systems as natural analogies of future oceans, and (2) the originality of our study. Following also your advice, our position is clear, this kind of systems, although not perfect, might be used with caution as natural analogies of future oceans.

About the first point, we have extensible addressed this question and provide solid and detailed evidence in comparison to previous studies. We have made changes on the text and added a discussion about the variability found at these natural systems, their limitations and usefulness, comparing our findings with others previous studies results. In this version more information is included in this sense.

In lines 260 - 277, we added examples of other acidified systems "These variations are very similar to other acidified natural systems. For example, the CO2 vent of Ischia (Italy) has pH levels from 6.07 to 8.17,  $\Omega$ aragonite from 0.07 to 4.28 and  $\Omega$ calcite from 0.11 to 6.40 (Hall-Spencer et al., 2008). The one from the island of Vulcano (Italy) has pH values between 6.80 and 8.20,  $\Omega$ aragonite from 1.49 to 4.65, and  $\Omega$ calcite from 2.28 to 7.00 (Boatta et al., 2013). Meanwhile the CO2 seeps from Papua New Guinea have pH levels between 7.29 - 7.98,  $\Omega$ aragonite 1.2 - 3.4 and  $\Omega$ calcite between 1.36 - 5.12 (Fabricius et al., 2011). That from Shikine island (Japan) have pH values between 6.80 and 8.10,  $\Omega$ aragonite from 0.20 to 2.22, and  $\Omega$ calcite from 0.30 to 3.45 (Agostini et al., 2015)".

And we also emphasized that these systems are not entirely perfect to predict the ocean future. We added "Although these systems are far from being perfect predictors of the ocean future due to their chemical variability and physical limitations, they have proven to be important tools for the study of ocean acidification (Foo et al., 2018; González-Delgado and Hernández, 2018; Aiuppa et al., 2020). These natural acidified systems, such as Punta de Fuencaliente system (PFS), can be used as natural analogue of climate change scenarios predicted by the IPCC (2014) (Fig. 6)."

In lines 282 – 292, we added some specificities of our system that make it different and we highlight that several caveats for future prediction experiments should be considered. "Moreover, the acidified system of La Palma highlight by the absence of bubbling, since the volcanic degasification takes place in the aquifers and not directly on the coast as in other acidified systems of volcanic origin (e.g. Hall-Spencer et al., 2008; Fabricius et al., 2011) (Fig. 5). This could give us new insights of the effect of acidification in situ avoiding the effects of bubbling (González-Delgado and Hernández, 2018). Nevertheless, several caveats for future prediction experiments should be considered, as well as in other natural acidified systems, especially those related with increased alkalinity values in the submarine discharge. First, there is a clear tidal influence, this is an important force that controls the acidified brackish water discharges. Although a fluctuation of the emission is observed, normal ocean conditions can occur for a short time, about 2-4 hours per day, during high tide, and depending on the oceanic conditions (Viotti et al., 2019). The pHT,is is severely affected by the location, reaching down to  $\sim 7.2$  in the emissions points, so a careful selection of the study sites is recommended, depending on the study objectives (Fig. 6). This tidal phenomenon has also been reported in other acidified natural systems such as Puerto Morelos in Mexico (Crook et al., 2012) and Ischia (Kerrison et al., 2011). However, the pH time fluctuation can be used to our advantage, as a daily and seasonal fluctuation of the pH is normal in coastal habitats environment (Hofmann et al., 2011). So, it could be considered very useful to incorporate pH variability in ocean acidification studies as environmental fluctuations that can have a large impact on marine organisms (Hofmann et al., 2011)."

In lines 302 - 304, we added the other studies should also be considered. "Therefore, measurements of heavy metals and other elements in seawater should be considered in following studies."

In relation to this first controversial point, we would like to clarify that the reviewer 2 mention previous literature that shows the cavities of using natural CO2 labs against our position, but we do also recognize these cavities in our manuscript. In lines 278 - 315, in addition to what is indicated above regarding this point, we have also added "The high concentration of bicarbonate in the brackish waters also implies an extra contribution of alkalinity and carbonate that can buffer the effect of acidification in the area, so it is necessary to take this into account when making predictions of the future. These values together with calcium content are especially important factors in the case of the saturation state for both calcite and aragonite, that shows high values for seawater with low pH values. Therefore, despite the fact that we are dealing with a subtropical ecosystem, the values obtained in both saturation states are more similar to the predictions for a tropical ecosystem, such as the values found in Papua New Guinea seeps (Fabricius et al., 2011; IPCC, 2014).

Finally, the area is not very large and only one type of rocky benthic habitat, the most typical community of Canary island, is present at the PFS (Sangil et al., 2018).

Therefore, all conclusions derived from this acidified system should be interpreted with caution and has local effects. Hence, it is crucial to establish a collaborative network of researchers who are working in other natural acidified systems worldwide to have a more realistic interpretation of future ocean scenarios."

However, her/his point for all these kinds of studies, done in natural acidified areas, is that "because the environmental conditions are not what we expect for the future and responses only confusing", we disagree with her/his interpretation.

In this sense, it could also be said that the studies carried out in the laboratory are not real either. In these experiments, generally considering only one biological species, one or two variables such as pH and T are controlled and the joint effects of the multiple stressors that affect biological communities are not taken into account. However, they are essential to be able to interpret more complex systems.

We believe that these studies are necessary to understand the ecosystem scale of the ocean acidification problem we are facing. Therefore, we expect that with this new version of the manuscript, we have better explained our argument and the importance of this place. We now support our argument, as you recommend, with a stronger discussion using previous literature.

About point 2, we have now included more information about our previous studies to have a better context of the present study and to highlight its novelty.

In lines 216 - 223, we added "Although CO2 emissions on Fuencaliente coast had already been detected (e.g. Hernández et al., 2016; Viotti et al., 2019), this is the first time that this naturally acidified system has been described chemically and physically. Previous works have focused on punctual questions; Hernández and collaborators (2016) published for the first time the presence of CO2 SGD in Fuencaliente, specifically in Las Cabras beach. Later, in the thesis by Pérez (2017), as well as in the conference papers by González-Delgado et al (2018a,b) and in the article by Viotti et al. (2019), new points of acidification were discovered on Playa del Faro and Los Porretos. However, in none of them was a chemical characterization of the whole area made as here." In this study, three of the four variables that define the CO2 system were measurements (pH, total alkalinity and total dissolved inorganic carbon) and CO2 certified reference material was used to collaborated our carbon dioxide system measurements.

However, we must say that our study is original, and the data have not been previously published, no doubts about that.

**Responses to referee#2 revision:**

**Comment 1:**"The Authors corrected most of the mistakes made in the original version and added some details which were lost. This help to better understand the work done. However, the authors did not consider reviewing their position regarding the ability of this new system to become a laboratory for understanding the effect of climate change on organisms. Personally, I find this position unusual. I perfectly know that some of the so called natural analogue found so far are far to be perfect, and that is the problem. Most of conclusions from some of the studies published, although always important to advance in science, make actually only noise..., because the environmental conditions are not what we expect for the future and responses only confusing."

**Response**: We presented an improved version of the manuscript thanks to yours and editor/reviewer's comments.

We understand that the increase in alkalinity makes the conclusion on this area different to what is expected in the future. However, not only in this region but also in upwelled corrosive environments such as California upwelling region (Feely et al., 2008; 2016), increase in acidity, inorganic carbon and normalized alkalinity to a constant salinity have been shown. Even with this difference, we consider we could clearly maintain our position about this kind of area confirming the differences, and although not a perfect system, it can be used, with this in mind, as natural analogues to future oceans but also to previous times. We have also, already proven the usefulness of this natural systems to demonstrate that while acidification may affect the calcification process of the mollusc Phorcus, it's not affecting its survival (Viotti et al., 2019 MERE) or can also indirectly benefice the growth of some calcifying invertebrates (Uthicke et al., 2019). These papers in natural environments, rather than adding noise and confusion, have helped to understand the complexity of ocean acidification phenomena. We believe that these works provide valuable information that complements the lab experiments and model predictions. Indirect ecosystem effects can only arise in these natural environments. We have added more information about this issue at the discussion section in lines 261 - 316 as indicated above in the comments to the editor. We have to recognize the limitations of using these spots and with doing so we are informing others on how to use the information gather at this spot. This is also an important point that makes this manuscript interesting for other OA researchers.

**Comment 2**: "Looking to the description of your system I see: - the lack of CO2 seeps. So the ms must be around submarine groundwater influence and the word CO2

**seeps should not be mentioned because they do not exist, or this study do not show if they are in the near- or offshore water."**

**Response:** We know that  $CO_2$  emissions are underground affecting the groundwater that is discharged into the coast. Although we do not directly analyse  $CO_2$  emissions, we comment additional data from a previously work written by Soler in 2007. In this study, Soler talks about the presence of CO2 seeps due to the remnant volcanic activity in the area. The presence of CO2 emanations in land, have also been previously found in the area (Padrón et al., 2015). In any case, we have now used the term submarine groundwater discharge (SGD) along the text.

We have better explained this issue in the manuscript for better understanding. In lines 221 - 227, we have added "Although CO2 emissions on Fuencaliente coast had already been detected (e.g. Hernández et al., 2016; Viotti et al., 2019), this is the first time that this naturally acidified system has been described chemically and physically. Previous works have focused on punctual questions; Hernández and collaborators (2016) published for the first time the presence of CO2 SGD in Fuencaliente, specifically in Las Cabras beach. Later, in the thesis by Pérez (2017), as well as in the conference papers by González-Delgado et al (2018a,b) and in the article by Viotti et al. (2019), new points of acidification were discovered on Playa del Faro and Los Porretos. However, in none of them was a chemical characterization of the whole area made as here. Our results reveal the continuous influence of brackish water discharge in the acidification process of Punta de Fuencaliente System (PFS), which had been missed before (Fig. 5). Similar to aerial remnant volcanic activity in La Palma that generates high CO2-diffusiveatmospheric concentration (Padrón et al., 2015), submarine remnant volcanic activity causes the acidification process found here, as indicated by the chemical composition of the groundwater analysed, which is less than 200 m from the coast (Soler, 2007). The activity of this SGD is comparable with other CO2 vent and seep systems worldwide (references within González-Delgado and Hernández, 2018)."

**Comment 3**: "- the quality of the groundwater discharge makes conditions too variable in pH (yes as most of the natural analogues), too altered (in bicarbonate, At, calcium, Mg, likely other metals etc), too limited in space (meters from the beach with a tide of 2 m!!). Ok, no problem, **nice and useful** but I **continue to think that it is not an exceptional spot to study OA**, not great to study **how life has persisted through past eras** (what a pretention!). It could help to understand the early life on Earth from the Precambrian..????? Could allow to disentangled adaptation and evolution...??? Please, do not speculate."

**Response:** We believe that this kind of comment is inappropriate and close to be consider an offence to our ideas and work. We agree with reviewer that we cannot study in this environment what has happened in the Precambrian or even in past eras, but the pH conditions you observe in the PFS are those indicated for those periods. However, this kind of extreme environments are clearly useful to study the type of organism that life there and their biology. In this sense, we can get closer to understand how marine organism may have lived in pass eras with "similar" conditions. It was not the aim of the present paper but the expected (future work to be monitored) high frequency change in acidification with tides makes this system appropriate for long time process studies, resilience, adaptation and evolution of species under the stress conditions imposed by both lunar cycles effects on tides and changing groundwater CO2 content. The ocean acidification process is not a constant process and seasonal, interannual and decadal trends have been observed in the long ocean time series such as BATS (Bates and Johnson, 2020). The area, presented hourly, daily, monthly and longer time variability affecting the live in the area or any organism we can set in the area to study its evolution. We consider, as it has been assumed by many previous workers in many other CO2 natural emitting sites included in the introduction section, that these places are very useful. We assume that using this type of systems to interpret, or at least to have an approximation, even if it is not perfect, can provide insight about past and future situations.

Of course, there are limitations, but there are limitations and assumptions in every experiment and model we design, in every decision we made to choose our hypothesis, this is the science we use today. You also mentioned adaptation and evolution. In this kind of natural experiment, we have found both adaptation and rapid evolution of seagrasses, algae and invertebrates (Kumar et al 2017; Olivé et al 2017; Harvey et al 2015), and this is interesting to disentangle a bit how evolution works. Why these authors have used this kind of experiments if their area do not perfectly match the projected future ocean conditions? Why we should *a priori* limit or prejudge the use of this areas? We do believe these are exceptional spots to study adaptation, evolution and how life deal with this unusual ocean conditions.

To better explain how important these areas for this king of adaptation/evolution studies are we have included below some concepts and the scientific context that, maybe, makes you see things from a different perspective. Species show phenotypic plasticity that may allow an organism to maximize fitness (Gotthard & Nylin, 1995) and adaptive phenotypic plasticity is expected to evolve in populations subject to contrasting environmental conditions (Via et al 1995). Intraspecific variability of this capacity could be then investigated to understand population resilience and adaptability on the time scale of actual scenarios, as the pH gradient presented in this manuscript. Some studies have already demonstrated variation in the degree of plasticity among populations in response to degree of variation in the environment (Kaitala et al., 1991; Leips & Travis, 1994; Morey & Renick, 2004). However, few studies have explored whether historical changes in environments or these kinds of gradients are associated with the evolution of phenotypic plasticity, local adaptations and time needed to evolve (Morey & Reznick 2004); and just one study has explored the advantages of these local adaptation modes to counteract ocean acidification (Kelly et al., 2013). Therefore, we do not speculate, we believe and have shown that these areas are exceptional spots to study adaptation and evolution.

Following editor recommendations, we have added more information and discussion to better explain the utility of these spots. For instances, we have compared our spot with the one in Ischia, Italy (Hall-Spencer et al., 2008), and the one in Papua New Guinea (Fabricius et al., 2011). We have added this discussion in lines 260 - 276. In general, even with noise (variability) associated, it has already been proven that the results obtained in these natural areas are important to better understand the effect of ocean acidification on an ecosystem level or to test adaptation and evolution.

**Comment 4**: - "the accurate chemical characterization" is a two date sampling on the seawater carbonate chemistry only. No mention about other true chemicals and metals discharged. IPCC does not include pollutants in its scenarios."

**Response**: Our data was obtained during three different months over two years, and we believe that this set of data can clearly be used to establish a chemical characterization of the carbonate system for the area. Of course, you can always have more data to characterize something, however, the limited funds we had did not allowed us to reach this point. It is true that it would have been great to provide, along with the water carbon system, data on metals and other elements but this has not been possible for the moment. We have added a clarification regarding this in lines 297 - 298.

**Comment 5**: "- the sampling was made, I insist, in the intertidal zone (i.e., The intertidal zone is the area where the ocean meets the land between high and low tides) when considering a tide of 2 m as at the study site. The effect of discharged water, so the low pH area, is close to the beach and immediately dissipate 50 m far, even less

depending the study site, from the shore. The lat & long scale does not help to appreciate the real extension of the affected area but having a look in google earth it is clear that for instance Playa del Faro (the biggest area of this study) is a 80m long."

**Response**: The sampling was made underwater using bootless in the subtidal area. Our study site is always cover by water, even in low tide. We also provide a picture, see bellow, to demonstrate to you that we were working underwater.

We believe that the scale is appreciated when you compare the graphs with Figure 1c, hence we have added the star-shaped mark that relates the interpolation graphs to the 1c map. The area is not very large, as it was already mentioned in the previous version of the manuscript (lines 313 - 315), this is one of the caveats of the spot, because we can only study the effect of the acidification on the local communities presents in the area. We have also wished to have a larger acidified areas with a proper temperature gradient to perfectly much the IPCC projections, but nature is capricious. We also know that many scientists would like to have a lab where species could be set and be studied in a tank of the dimension of this beach with gradients in pH. This is what this natural place offers.

**Comment 6**: "- This ms is only about the seawater conditions found. There is no data about what could be the biota living and affected by the gradient in the beach, which is an area affected by the tide and the local strong waves. It is not enough to cite an in press paper (not cited in the literature list). All discussions on that part are purely speculations."

**Response:** Yes, you are right the manuscript is about the chemical characterization of an especial spot in La Palma island but not about the present biota. However, we have already published a paper about the biota and we are now working with new collected data that we would like to comment. We are sharing our observations and we have just included a comment for the readers to better visualize the study area. We would like to keep this small, "naturalistic", comment.

**Comment 7**: "- Fig. 5 is not really important and good for a book. How brackish groundwater is released and reach the sea is an already known mechanism. - Fig. 6 is honestly inacceptable. - Fig. B1. I do not understand what exactly means.

**Response:** We believe that Figure 5 is adequate to understand the phenomenon that occurs in the coast of Fuencaliente, not only to show "*How brackish groundwater is released and reach the sea*" but also, how CO2 emissions mixed with this brackish groundwater. Figure 6 shows the pH levels found in the spot area and clearly shows what you were asking in your comment 5, and can be related with those predicted pH values for the future by the IPCC. We would like to keep figure 6. Figure 1B is a complement to the description of the study area, this figure not only summarizes the locations, months and parameters sampled at each site, but also shows the location of the lava flows from the last eruption that took place in the area.

**Comment 8**: "Why min pH was used? Why the data are connected by a line although sampling is not continuous? Clearly, tide has a direct effect of the groundwater release"

**Response:** We have modified this graph to better represent the data. We agree with you that the tide has a direct effect on the release of groundwater. This is one of the findings of this study. The effect of the tide should not be neglected in any hydrothermal system near the coast, as demonstrated in Santana-Casiano et al., 2016. In this work (figure 3 and 4) it is demonstrated how the emissions from the Tagoro volcano, its crater at 88 m depth, are affected by the tides.

**References**

Bates, N. R. and Johnson, R. J. Acceleration of ocean warming, salinification, deoxygenation and acidification in the surface subtropical North Atlantic Ocean. Communications Earth & Environment, 1(1), 1-12, 2020.

Fabricius, K. E., Langdon, C., Uthicke, S., Humphrey, C., Noonan, S., De'ath, G., Okazaki, R. ,Muehllehner, N., Glas, M. S. and Lough, J. M.. Losers and winners in coral reefs acclimatized to elevated carbon dioxide concentrations. Nat. Clim. Chang. 1 (3), 165–169, 2011.

Feely, R. A., Sabine, C. L., Hernandez-Ayon, J. M., Ianson, D., Hales, B. Evidence for upwelling of corrosive "acidified" water onto the Continental Shelf. Science 320 (5882), 1490e1492, 2008 http://dx.doi.org/10.1126/science.1155676.

Gotthard K. and Nylin S. Adaptive plasticity and plasticity as an adaptation: a selective review of plasticity in animal morphology and life history. Oikos. 74:3–17, 1995.

Hall-Spencer, J. M., Rodolfo-Metalpa, R., Martin, S., Ransome, E., Fine, M., Turner, S. M., Rowley, S. J., Tedesco, D. and Buia, M. C. Volcanic carbon dioxide vents show ecosystem effects of ocean acidification. Nature 454 (7200), 96–99, 2008.

Harvey, B. P., McKeown, N. J., Rastrick, S. P., Bertolini, C., Foggo, A., Graham, H., Hall-Spencer, J. M., Milazzo, M., Shaw, P. W., Small, D. P. and Moore, P. J. Individual and population-level responses to ocean acidification. Scientific reports, 6, p.20194, 2016.

Kaitala A. Phenotypic plasticity in reproductive behaviour of waterstriders: trade-offs between reproduction and longevity during food stress. Funct. Ecol. 5:12–18, 1991.

Kelly M. W., Padilla-Gamiño J. L., Hofmann G. E. Natural variations and the capacity to adapt to ocean acidification in the keystone sea urchin *Strongylocentrotus purpuratus*. Glob. Chang. Biol. 19: 2536-2546, 2003.

Kumar, A., Castellano, I., Patti, F. P., Delledonne, M., Abdelgawad, H., Beemster, G. T., Asard, H., Palumbo, A. and Buia, M. C. Molecular response of Sargassum vulgare to acidification at volcanic CO 2 vents: Insights from de novo transcriptomic analysis. Molecular ecology, 26(8), pp.2276-2290, 2017.

Leips J., Travis J. Metamorphic responses to changing food levels in two species of hylid frogs. Ecology. 75:1345–1356, 1994.

Morey S. R., Reznick D. N. The relationship between habitat permanence and larval development in California spadefoot toads: field and laboratory comparisons of developmental plasticity. Oikos. 104:172–190, 2004.

Olivé, I., Silva, J., Lauritano, C., Costa, M. M., Ruocco, M., Procaccini, G., Santos, R. Linking gene expression to productivity to unravel long-and short-term responses of seagrasses exposed to CO2 in volcanic vents. Sci. Rep. 7, 42278, 2017.

Padrón, E., Pérez, N. M., Rodríguez, F., Melián, G., Hernández, P. A., Sumino, H., Padilla, G., Barrancos, J., Dionis, S., Notsu, K. and Calvo, D. Dynamics of carbon dioxide emissions from Cumbre Vieja volcano, La Palma, Canary Islands. Bull. Volcanol., 77, 1-15, https://doi.org/10.1007/s00445-015-0914-2, 2015.

Soler-Liceras, C. La historia de la Fuente Santa. Editorial Turquesa. Santa Cruz de Tenerife, 2007.

Uthicke, S., Deshpande, N. P., Liddy, M., Patel, F., Lamare, M., and Wilkins, M. R. Little evidence of adaptation potential to ocean acidification in sea urchins living in "Future Ocean" conditions at a CO2 vent. Ecol. Evol., 9(17), 10004-10016, 2019.

Via S., Gomulkiewicz R., De Jong G., Scheiner S. M., Schlichting C. D., van Tienderen P. H. Adaptive phenotypic plasticity: consensus and controversy. Trends Ecol. Evol. 10: 212–217, 1995.

Viotti, S., Sangil, C., Hernández, C. A. and Hernández, J. C. Effects of long-term exposure to reduced pH conditions on the shell and survival of an intertidal gastropod. Mar. Environ. Res., 152, p.104789, https://doi.org/10.1016/j.marenvres.2019.104789, 2019.